# SWE-Search: Enhancing Software Agents with Monte Carlo Tree Search and Iterative Refinement

**Antonis Antoniades**[1*]**, Albert Örwall**[2*]
**Kexun Zhang**[3]**, Yuxi Xie**[4]**, Anirudh Goyal**[5]**, William Wang**[1]
[1]University of California, Santa Barbara, [2]Moatless AI, [3]Carnegie Mellon University,
[4]National University of Singapore, [5]Mila

## Abstract

Software engineers operating in complex and dynamic environments must continuously adapt to evolving requirements, learn iteratively from experience, and reconsider their approaches based on new insights. However, current large language model (LLM)-based software agents often follow linear, sequential processes that prevent backtracking and exploration of alternative solutions, limiting their ability to rethink their strategies when initial approaches prove ineffective. To address these challenges, we propose SWE-Search, a multi-agent framework that integrates Monte Carlo Tree Search (MCTS) with a self-improvement mechanism to enhance software agents' performance on repository-level software tasks. SWE-Search extends traditional MCTS by incorporating a hybrid value function that leverages LLMs for both numerical value estimation and qualitative evaluation. This enables self-feedback loops where agents iteratively refine their strategies based on both quantitative numerical evaluations and qualitative natural language assessments of pursued trajectories. The framework includes a SWE-Agent for adaptive exploration, a Value Agent for iterative feedback, and a Discriminator Agent that facilitates multi-agent debate for collaborative decision-making. Applied to the SWE-bench benchmark, our approach demonstrates a 23% relative improvement in performance across five models compared to standard open-source agents without MCTS. Our analysis reveals how performance scales with increased inference-time compute through deeper search, providing a pathway to improve software agents without requiring larger models or additional training data. This highlights the potential of self-evaluation driven search techniques in complex software engineering environments.

## 1 Introduction

Software engineering is a complex and iterative process involving exploration, problem-solving, and decision-making under uncertainty. Tasks such as debugging, feature development, and code refactoring require continuous assessment of different approaches, frequent backtracking, and the incorporation of new information. While machine learning has made progress in automating parts of this workflow (Li et al., 2022; OpenAI et al., 2024; Ouyang et al., 2022; Yang et al., 2024b), replicating the adaptive and strategic behavior of human engineers remains a significant challenge. This is due to the inherently non-linear and iterative nature of software engineering, where engineers dynamically explore various solutions, refine strategies based on feedback, and collaborate to identify the most effective path forward. Current large language model (LLM)-based software agents (Xia et al., 2024; Zhang et al., 2024d), while powerful, often struggle with complex, long-horizon tasks that require adaptive strategies and flexible reassessment over time. These agents can become trapped in repetitive patterns, limiting their effectiveness in tackling more intricate software engineering problems.

---

*Denotes equal contribution. **Correspondence to:** antonis@ucsb.edu, albert@moatless.ai.
**Code:** github.com/aorwall/moatless-tree-search, **Demo:** https://streamlit.moatless.ai

To address these challenges, we introduce **SWE-Search**, a multi-agent system that replicates the adaptability, iterative learning, and collaborative decision-making of human engineers. SWE-Search is designed to address three critical needs in software engineering:

*Flexible Exploration and Adaptation*: Engineering problems often require exploring multiple approaches and adapting strategies based on evolving information (Li et al., 2022). SWE-Search's SWE-Agent operates in a flexible state space, allowing it to fluidly transition between actions such as planning, searching, and editing. This design mirrors the way engineers backtrack and adjust their approach dynamically, ensuring the agent can revise its course when faced with new challenges or information, and points towards the direction of more general, open-ended systems (Wang et al., 2023; Ma et al., 2024a; Lu et al., 2024b; Faldor et al., 2024; Hu et al., 2024; Lu et al., 2024a).

*Iterative Learning through Feedback*: Effective engineering relies heavily on continuous testing and refinement. To replicate this, SWE-Search integrates a Monte Carlo Tree Search (MCTS) (Silver et al., 2016b) planning module paired with a Value Agent. The MCTS module balances exploration and exploitation to guide the agent through complex solution spaces. The Value Agent augments this process by providing both utility estimates and qualitative feedback, allowing the agent to iteratively improve its decision-making based on past experiences, similar to how engineers refine their work through feedback and debugging.

*Collaborative Decision-Making*: Complex problems often benefit from diverse perspectives (Khan et al., 2024; Amayuelas et al., 2024; Du et al., 2023; Zhang et al., 2024c). In SWE-Search, once a set of potential solutions is generated, the Discriminator Agent facilitates a multi-agent debate. Each agent advocates for different solutions by presenting arguments, which are critically evaluated by a judge agent. This process mirrors real-world engineering collaboration, where teams deliberate to refine and select the most robust solutions.

We evaluate SWE-Search on the SWE-bench-lite, a benchmark which tests agents' ability to resolve real-world repository-level issues by generating code patches that fix failing tests. SWE-Search demonstrates a 23% relative performance improvement across five models compared to standard open-source agents. We explore how performance scales with increased search depth and identify key factors that enhance self-assessment in software agents. Our work demonstrates the potential of MCTS and iterative learning to improve agent reasoning and planning in dynamic, complex domains like software engineering, introducing a new paradigm for autonomous software development.

## 2 RELATED WORK

**Search methods** Various search approaches have been applied to Large Language Models (LLMs) to facilitate System 2 (Kahneman, 2011; Saha et al., 2024; Pan et al., 2023; Bounsi et al., 2024) thinking in non-linear reasoning structures. A critical feature of these approaches is their ability to backtrack. Unlike greedy processes (Black, 2005), search algorithms explore multiple branches at each step, potentially escaping paths that lead to dead ends. These methods differ in their strategies for exploring and memorizing possible choices, and in their heuristics for switching between them. Breadth-first search (Moore, 1959) maintains all possible search paths, incurring significant memory and computational costs. Depth-first search (Cormen et al., 2009), in contrast, prioritizes the most promising path in a more greedy manner. When applied to LLMs, these methods demonstrate a trade-off between diversity and quality in text generation (Yao et al., 2023). The A$^*$ algorithm (Hart et al., 1968) combines aspects of breadth-first and greedy search to find optimal solutions using a predetermined evaluation function. In this work, we adopt Monte Carlo Tree Search (MCTS) (Coulom, 2007), an advanced search algorithm that conducts statistical tree search without requiring dedicated evaluation heuristics for each state. MCTS has achieved impressive results in complex strategy games (Silver et al., 2016b), protein folding (Jumper et al., 2021), and algorithm discovery (Fawzi et al., 2022).

**Software Agents** Software agents are designed to perform autonomous actions within large codebases. Given a repository-level task, these agents typically locate relevant files and code segments before implementing necessary changes. We focus on the SWE-bench task (Jimenez et al., 2024), which involves resolving real-world GitHub issues. Among the agents with disclosed technical details on SWE-bench, Yang et al. (2024b) introduced the concept of agent-computer interfaces with SWE-agent. OpenDevin (Wang et al., 2024b) presents a collection of community-driven agents,

including CodeAct (Wang et al., 2024a). The Agentless approach demonstrated competitive performance using a simple two-step process of localization and repair. AutoCodeRover (Zhang et al., 2024d) incorporated advanced code tools such as abstract syntax trees and spectrum-based fault localization. The Alibaba Lingma Agent (Ma et al., 2024b) introduced a search-based approach for repository exploration, followed by a structured editing phase. While effective, it constitutes a more hand-designed solution specifically designed to interface with the search functionality of their agent. Finally, Brown et al. (2024) show that repeated trajectory sampling using the exact same agent/model setup can yield results with high variance.

## 3 METHODOLOGY

SWE-Search is a multi-agent system designed to tackle complex software engineering tasks by integrating dynamic planning, value estimation, and deliberative decision-making. The core motivation behind this method is to emulate the sophisticated, iterative workflows of human software engineers, where exploration, planning, and collaboration are crucial to solving intricate problems. By leveraging the strengths of Monte Carlo Tree Search (MCTS) for planning, a Value Agent for utility estimation and feedback, and a Discriminator Agent for final decision-making through debate, SWE-Search provides a comprehensive, adaptive framework capable of navigating and solving real-world software engineering challenges.

SWE-Search consists of four primary components that work in synergy:

*SWE-Search Framework and Action Agent*: Building on the moatless-tools framework (Örwall, 2024), SWE-Search operates in a dynamic code environment with a flexible state-space and a git-like commit tree structure. This design facilitates efficient backtracking to previous states, enabling the Action Agent to explore diverse solution trajectories. The adaptable state-space enhances the system's ability to exploit the MCTS module effectively.

*Value (Function) Agent*: To approximate the utility of each observation, we employ an LLM-based value function, which in addition to outputting a value, also generates an explanation in natural language. This explanation can be leveraged to improve subsequent actions from parent nodes, enabling iterative self-improvement of the search process.

*Search Algorithm*: The core of SWE-Search's exploration strategy is based on a Monte Carlo Tree Search (MCTS) which uses a heuristic-based selection process similar to AlphaZero (Silver et al., 2016a), specifically tailored for software engineering tasks. This modified MCTS algorithm effectively balances exploration and exploitation, helping the agent explore a diverse set of solutions and converge quickly on the most promising strategies.

*Discriminator Agent*: In the final stage of SWE-Search, the Discriminator Agent evaluates the solutions generated by the search process. Inspired by multi-agent debate frameworks (Du et al., 2023; Khan et al., 2024; Amayuelas et al., 2024), this agent engages in a structured debate, where multiple agents argue for or against the proposed solutions. The debate process not only surfaces diverse perspectives but also leads to a more rigorously justified final decision.

### 3.1 PROBLEM FORMULATION

The task of the SWE agent can be formalized as a tuple $\mathcal{M} = (\mathcal{S}, \mathcal{C}, \mathcal{A}, \mathcal{V}, \mathcal{P}, p_0, \rho)$. Here, $\mathcal{S}$ represents the state space, encompassing all possible states such as the current context of the files the agent is working on and the overall status of the codebase. The context space, denoted as $\mathcal{C}$, includes metadata about the repository and the initial problem description. The value function $\mathcal{V}$ assigns a utility score to each state-action pair $O(a, t)$, guiding the agent's decisions.

The environment's dynamics are defined by a context-dependent transition function $\mathcal{P} : \mathcal{S} \times \mathcal{A} \times \mathcal{C} \to \Delta(\mathcal{S})$, which models the evolution of the repository's state after each action. The initial state distribution, $p_0 : \mathcal{C} \to \Delta(\mathcal{S})$, specifies how the initial state depends on the given context, while $\rho \in \Delta(\mathcal{C})$ defines the distribution over contexts.

Given an initial context $c \sim \rho$ and an initial state $s_0 \sim p_0(\cdot \mid c)$, the SWE agent executes its policy $\pi : \mathcal{S} \times \mathcal{C} \to \Delta(\mathcal{A})$, which selects actions based on the current state and context. At each time step $t$, the agent takes an action of type $\tau$, $a_{t,\tau} \sim \pi(s_{t,\tau}, c)$ and receives a corresponding reward

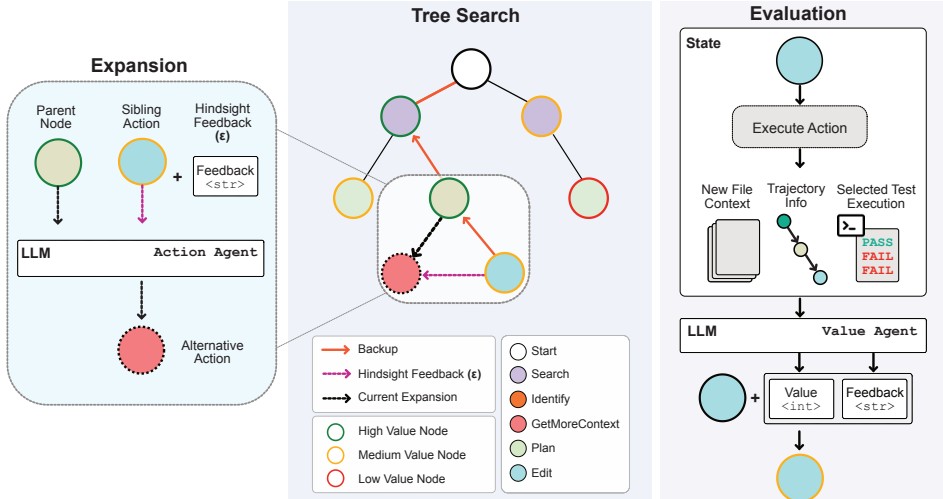

Figure 1: **SWE-Search Overview.** *Tree search.* Each state is represented as a node from which the agent can expand from, and each corresponding action is presented as an edge. *Evaluation.* Uses all relevant context including trajectory information, file context, and executed tests, to provide a quantitative value estimation and qualitative explanation in natural language. *Expansion.* Nodes can be expanded using value function feedback from future actions.

$\mathcal{R}(s_{t,\tau}, a_{t,\tau}, c)$. The environment then transitions to a new state $s_{t+1} \sim \mathcal{P}(\cdot \mid s_{t,\tau}, a_{t,\tau}, c)$, and the agent continues to observe this updated state. Over time, this process generates a trajectory $\tau := \{s_{t,\tau}, a_{t,\tau}, r_t\}_{t=0}^{T}$ as the agent interacts with the environment.

The agent's objective is to maximize the cumulative reward over the trajectory, which is captured by the value function $v(s_{t,\tau}, a_{t,\tau}, \{s_i\}_{i=0}^{t-1}, \{a_i\}_{i=0}^{t-1})$. This value function depends not only on the current state and action but also on the history of previous states and actions, which deviates from the assumptions of a Markovian process. Formally, the agent seeks to maximize the expected cumulative reward, defined as: $\max_\pi V_T(\rho) = \max_\pi \mathbb{E}_\tau \left[ \sum_{t=0}^{T} R(s_{t,\tau}, a_{t,\tau}, c) \mid c \sim \rho; \pi \right]$.

This optimization captures the agent's (in-context) process, as it adjusts its policy $\pi$ to achieve the highest expected return across multiple trajectories, considering both current and historical information.

## 3.2 SWE-Search Framework and Action Agent

The SWE-Search Action Agent builds on the moatless-tools framework (Örwall, 2024). Its action space, $\mathcal{A}$, is organized as a two-tier hierarchy, comprising both action types and their corresponding specific actions. Formally, this can be expressed as $\mathcal{A} = (t, \tau, a) \mid t \in \mathcal{T}, a \in \mathcal{A}_t$, where $\mathcal{T}$ represents the set of action types (e.g., Search, Plan, Edit), and $\mathcal{A}_t$ is the set of possible actions corresponding to each type $\tau$. These actions range from tool invocations and code modifications to the generation of structured text. When an action is executed, the Agent transitions to a state of the corresponding action type $s_{t+1,\tau}$. To enhance the base agent's effectiveness in search-driven tasks, we introduced the following modifications:

One key modification we implemented is the expansion of the Plan state, allowing it to transition flexibly to any other state, rather than being limited to transitioning only to Edit. This change is motivated by the need to enable more dynamic and adaptive problem-solving behaviors within the agent. In the context of software engineering, rigid state transitions can be overly restrictive. For instance, during code modification tasks, an agent might recognize mid-process that further planning, additional searches, or different types of analysis are necessary before proceeding with edits. Restricting transitions only to editing would artificially constrain the agent, potentially leading it to suboptimal actions or causing it to become stuck in unproductive loops. By allowing transitions to any state, we empower the agent to adapt to new information as it arises (**Fig. 2**), exploring

a wider variety of trajectories. This enhanced flexibility reflects the iterative and often non-linear nature of real software engineering workflows, where engineers frequently revisit planning, testing, and research phases before committing to edits.

Second, the agent is empowered to execute any tests within the codebase at its discretion, as well as to create and implement new tests. The results of these tests are incorporated into both the value function and the agent's subsequent decision-making process. It is crucial to highlight that the tests required to resolve a given instance (i.e., fail-to-pass tests) are not explicitly revealed to the agent. However, the agent can leverage any pre-existing tests within the repository, simulating the behavior of a real-world software engineer. [1] We refer to this enhanced base agent as Moatless-Adapted.

### 3.3 VALUE (FUNCTION) AGENT

The role of the Value Agent extends beyond simply estimating the expected utility of a given state-action pair $V(s_t, a_t)$. In addition to calculating the value $v_n$, the Value Agent generates a written explanation, denoted as $\varepsilon$. This explanation serves a dual purpose: it provides transparency into the decision-making process and functions as feedback for the Action Agent, which can leverage this explanation when re-expanding from the parent node of $O_n$ (see **Figure 1**, *hindsight feedback*). This approach enables the system to iteratively refine its decision-making process, mirroring how a human software engineer continuously re-evaluates their approach based on new information to improve their problem-solving strategy.

The input to the value function consists of all state-action pairs up to and including the current state being evaluated, alongside specific instructions on how to assess the state. This allows the Value Agent to contextualize the decision within the trajectory, accounting for the sequence of actions and states leading up to the present. The final output of the value function can be formalized as:

$$(v_t, \varepsilon_t) = V(s_{t,\tau}, a_{t,\tau}, \{s_i\}_{i=0...t-1}, \{a_i\}_{i=0...t-1}) \tag{1}$$

Here, $v_t$ represents the expected utility of the current state-action pair, while $\varepsilon_t$ is the accompanying explanation.

In practice, the Value Agent is tasked with analyzing the entire trajectory leading up to the current state-action pair, providing not only the required utility estimate $v_t$, but also a detailed explanation $\varepsilon_t$. This explanation is critical for the agent's overall performance, as it offers insight into the reasoning behind utility estimates, which in turn informs the Action Agent's future decisions. We have observed that one of the key factors driving the effectiveness of the Value Agent lies in the clarity and specificity of these explanations. A well-articulated explanation can illuminate the strengths and limitations of different state types (e.g., `Search`, `Edit`, `Plan`), helping the Action Agent better understand which types of states are more promising or risky to pursue.

By providing detailed feedback on the potential utility of different actions and contextualizing them within the broader trajectory, the Value Agent enables more informed and strategic decision-making by the Action Agent. This integration of both quantitative and qualitative feedback leads to improved performance and more adaptive behavior throughout the task (**Fig. 4a**).

### 3.4 SEARCH ALGORITHM

Our search tree is structured with nodes representing states $\mathcal{S}_{t,\tau}$ and edges representing actions $\mathcal{A}_{t,\tau}$. The search algorithm employed is a modified Monte Carlo Tree Search (MCTS), specifically adapted for the tasks of the SWE-Agent. Unlike prior approaches for web agents that utilize language models in the selection process (Koh et al., 2024; Zhang et al., 2024b), we deliberately choose not to rely on language models for node selection. Instead, we adopt a more straightforward heuristic-based selection function, similar to the approach used in AlphaZero (Silver et al., 2016a; 2018). This decision is driven by the need for interpretability, efficiency, and the focus on tasks where heuristic-based exploration suffices to guide the agent effectively through complex software engineering environments.

---

[1]This approach aligns with the practices of other SWE agents, and has been validated by the authors of SWE-bench, who confirmed its legitimacy as long as the fail-to-pass tests remain concealed from the model.

At the core of our algorithm is a modified Upper Confidence Bound for Trees (UCT) selection criterion (Kocsis & Szepesvári, 2006), which determines the next node to expand. This criterion balances exploitation of known high-reward actions with exploration of less-visited states. We introduce additional terms to encourage strategic exploration early in the search process, and to penalize over-exploration at later stages when convergence on the optimal solution is desired. The modified UCT function is expressed as:

$$UCT(s, a) = exploitation + exploration + early\_depth\_bonus - late\_depth\_penalty \quad (2)$$

This can be expressed more formally as:

$$UCT(s, a) = V(s, a) + C\sqrt{\frac{\ln N(s)}{N(s, a)}} + \alpha e^{-\beta(d-1)} - \gamma\sqrt{d} \quad (3)$$

$V(s, a)$ is the value estimate of the state-action pair , $N(s, a)$ is the number of times the state-action pair $(s, a)$ has been visited, $N(s)$ is the visit count of state $s$, $d$ is the depth of the node in the search tree, and $C, \alpha, \beta$, and $\gamma$ are constants that control the balance between exploration, exploitation, and depth-dependent rewards and penalties.

This formulation is inspired by the way software engineers explore potential solutions to a task. In practice, an engineer's search process can be broken down into the following key phases, which our algorithm mirrors:

*Early Exploration*: Initially, an engineer explores a wide variety of potential approaches to fully understand the problem and identify promising strategies. This is encouraged in our algorithm by the $early\_depth\_bonus$, represented by the term $\alpha e^{-\beta(d-1)}$, which rewards exploration at shallow depths, simulating the early phases of wide exploration.

*Convergence and Exploitation*: As the engineer gains more information and narrows down the options, the focus shifts to exploiting the most effective solution paths. This transition is handled by the standard UCT exploitation term $V(s, a)$ and is further reinforced by the $late\_depth\_penalty$ $(-\gamma\sqrt{d})$, which discourages over-exploration as the agent delves deeper into the search tree.

*Quick Abandonment of Poor Strategies*: Software engineers are also adept at abandoning poor strategies when new information indicates that a particular approach is not viable. We capture this behavior by implementing a simple heuristic rule that abandons nodes associated with consecutive low rewards, ensuring that the agent does not waste resources on unproductive trajectories.

At each step, the node with the highest UCT value is selected for expansion, formalized as:

$$s^* = \arg\max_{(s,a)} UCT(s, a) \quad (4)$$

This approach effectively mimics the decision-making process of a software engineer, who balances exploration of potential strategies with a focus on converging towards the optimal solution, while remaining flexible enough to backtrack when necessary. By incorporating heuristic feedback and depth-based adjustments, the algorithm avoids getting stuck in unproductive paths and enhances the agent's ability to identify high-reward strategies with minimal computational overhead **Appendix 6**.

## 3.5 DISCRIMINATOR AGENT

The final stage of SWE-Search involves the Discriminator Agent, whose role is to evaluate the candidate solutions generated by the search process and select the one most likely to resolve the issue at hand. This module accepts up to five final solutions produced by the search and engages in a multi-agent debate to determine the most promising option. Drawing inspiration from recent work on persuasive multi-agent debates (Khan et al., 2024; Amayuelas et al., 2024), the Discriminator leverages the collective reasoning of multiple agents to ensure a more robust final selection. The complete prompts used in both the debate rounds and conclusion phase are provided in **Appendix L**. Configuration and hyperparameter details can be found in **Table 2**.

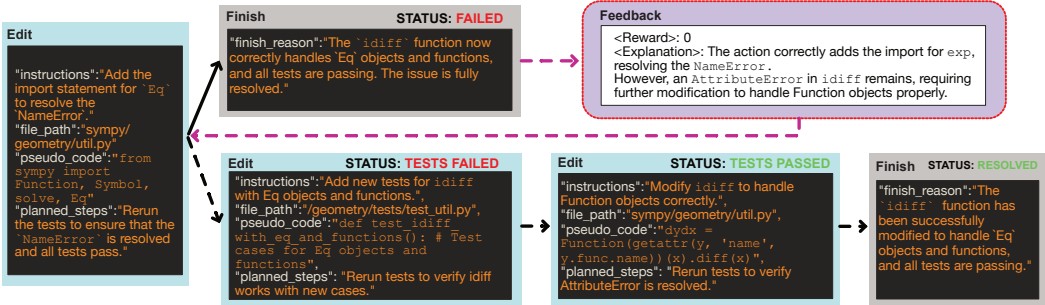

Figure 2: **Hindsight feedback error correction.** Instance sympy__sympy-15678, SWE-Search with Qwen2.5-72B-Instruct. Initially, the Action Agent performs edits and runs tests, which pass. It prematurely concludes the search. Without actually knowing the proposed solution does not resolve the issue, the Value Agent identifies potentially missed tests and assigns a low reward. Upon re-expansion using the Value Agent's feedback, new tests fail, prompting the Action Agent to make additional edits, which result in a preferred solution which ultimately resolves the issue.

The discriminator stage extends the MCTS framework by enabling direct comparison between candidate solutions through structured debate. While our value function achieves 73% accuracy in solution evaluation, empirical results show that the discriminator improves final selection accuracy to 84% (**Fig. 8a**). This improvement is particularly notable for cases where multiple solutions have similar value estimates but differ in implementation details. The structured comparison helps identify subtle trade-offs that may not be captured by the value function alone.

## 4 EXPERIMENTS

**Benchmark** For our experiments, we utilize SWE-bench Lite, a curated subset of the official SWE-bench, containing 300 instances. This dataset is specifically designed to be self-contained and focuses primarily on evaluating functional bug fixes, providing a controlled environment to assess the performance of our system.

**Evaluation Metrics** We use two metrics: resolve rate (*Pass@1*) and *Pass@5*. Resolve rate is the percentage of issues successfully resolved, measuring overall effectiveness. Pass@5 is the percentage of issues where a correct solution is found within five attempts. This allows us to assess the efficiency of the search in identifying successful bug fixes within a limited number of iterations.

**Baselines** Software agents leverage diverse tools, architectures, and models, leading to variability in their performance on subsets of the SWE-bench Lite dataset (Zhang et al., 2024a). For comparison, we build upon the moatless-tools framework (Örwall, 2024), a high-performing open-source agent commonly used in research settings (Chowdhury et al., 2024). To isolate the impact of our search approach, we adapt moatless-tools v0.0.2 as our baseline, referred to as Moatless-Adapted. This allows us to fairly compare the performance of SWE-Search against Moatless-Adapted across various models, including two closed-source models (GPT-4o, GPT-4o-mini) and three open-source models (Qwen2.5-72B-Instruct (Yang et al., 2024a), Llama-3.1-70B-Instruct (Dubey et al., 2024), and DeepSeek-V2.5 (DeepSeek-AI et al., 2024)). We also reference official moatless-tools GPT-4o results on SWE-bench Lite to ensure a fair and consistent comparison.

**Implementation Details** For consistency, we use identical prompts across all models. In SWE-Search, we limit each node to a maximum of three expansions and cap the total search iterations at 100. Further details on model hyperparameters can be found in **Appendix, 2**.

Table 1: Resolve Rate Comparison, SWE-bench Lite

| Model | Moatless-v0.0.2 | Moatless-Adapted | SWE-Search | % Δ |
|---|---|---|---|---|
| GPT-4o | 24.3 | 25.7 | 31.0 | +17 |
| GPT-4o-mini | – | 13.0 | 17.0 | +24 |
| Qwen-2.5-72b-Instruct | – | 18.0 | 24.7 | +27 |
| Deepseek-V2.5 | – | 16.3 | 21.0 | +22 |
| Llama-3.1-70b-Instruct | – | 13.6 | 17.7 | +23 |
| **Mean % Δ** | | | | **+23** |

## 4.1 EXPERIMENTAL RESULTS

### 4.1.1 SWE-SEARCH SURPASSES ALL CORRESPONDING BASE AGENTS AND ENABLES SMALLER, OPEN SOURCE MODELS TO APPROACH GPT-4O

On average, SWE-Search outperforms the baseline agent across all five models, achieving a 23% relative improvement (**Table 1**). Notably, SWE-Search with Qwen-2.5-72B-Instruct exceeds the performance of GPT-4o using the original Moatless-v0.0.2 framework, and closely matches its performance when compared with the Moatless-Adapted agent, with only a slight difference ($\Delta = -1\%$). Interestingly, all five models demonstrate significant improvement when utilizing the proposed approach, with consistent gains across different models.

### 4.1.2 SEARCH ENABLES AGENTS TO MAKE BETTER USE OF MORE FLEXIBILITY

To prevent goal divergence, most agents, including moatless-tools, rely on strict transition rules, where state transitions follow predetermined sequences (e.g., Search → Identify, Plan → Edit). In Moatless-Adapted, we introduce a more flexible transition logic that allows a Plan state to transition into any other state type. This added flexibility has both advantages and drawbacks. On the positive side, it enables the agent to autonomously correct its trajectory without external feedback, particularly when the necessary adjustments span only a limited portion of the task. However, this increased flexibility also introduces the risk of the agent becoming trapped in infinite loops. Without a high-level control mechanism to detect and mitigate these situations, the agent may fail to recover from such loops. This trade-off is evident in the modest performance difference between Moatless-v0.0.2 and Moatless-Adapted, with a slight performance improvement of only 1.4% (**Table 1**).

### 4.1.3 IMPACT OF HINDSIGHT FEEDBACK ON AGENT PERFORMANCE

One key advantage of utilizing LLMs for value estimation is their dual ability to provide both quantitative value estimates and qualitative assessments in natural language. These qualitative insights can significantly enhance the agent's action generation and search process by offering detailed feedback on potential errors or overlooked aspects of the task. In practice, feedback was also crucial in eliciting diversity in the actions taken by the agent, as without it, the agent would often take very similar actions when re-expanding from a parent node.

As shown in **Figure 2**, this mechanism plays a critical role in improving the agent's performance. During the initial expansion, the agent prematurely concludes that the task is complete. However, the value function correctly identifies gaps in the test coverage, specifically in addressing potential corner cases, and assigns a low reward. This feedback prompts the agent to re-expand the parent state, leading to the introduction of new tests, which subsequently fail. The agent then performs a series of edits (summarized in the figure for brevity), ultimately resolving the task correctly. Empirically, we observe that the instances unresolved by Moatless-Adapted but successfully solved by SWE-Search are often attributed to this search-and-feedback loop, where iterative feedback drives the agent toward a correct solution.

## 4.2 IMPORTANCE OF COMPREHENSIVE STATE INFORMATION FOR VALUE FUNCTION PERFORMANCE

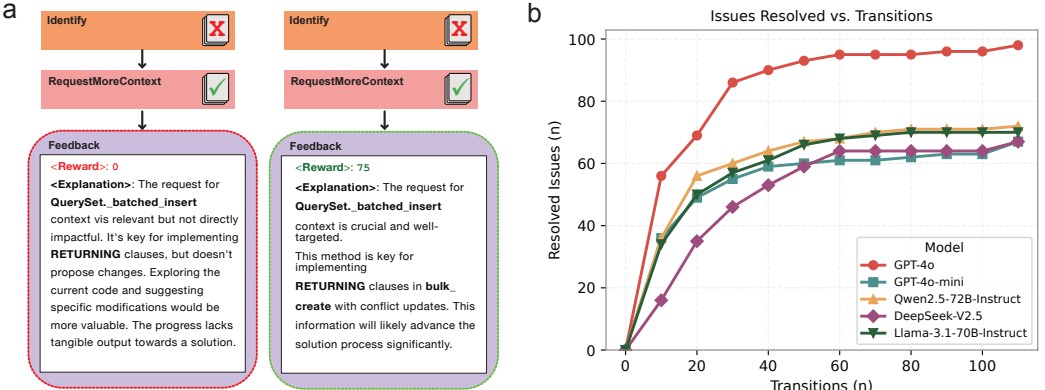

Figure 4: **(a) Importance of state-specific value prompts.** On the left and right are the respective Value Agents' outputs with and without state-specific prompts. While the action in both cases is effective in finding the right file, the non-state-specific scenario does not recognize this and assigns a low reward. On the contrary, the state-specific prompt correctly assigns a high reward to this state. **(b) Performance scaling with search depth across different language models.** The graph shows the number of issues resolved as a function of the number of transitions (search iterations) for all models used.

The effectiveness of SWE-Search hinges on the value function's ability to accurately differentiate between desirable and undesirable states, and to provide actionable feedback that drives improvement. However, our experiments revealed that the value function sometimes failed to recognize critical decision points in the search tree. It frequently misinterpreted the purpose of certain actions, leading to the undervaluation of effective strategies by assigning low rewards. As shown in **Figure 4a**, before the introduction of state-specific value prompts, the agent consistently assigned

| Model | Pass@1 | Pass@5 |
|---|---|---|
| GPT-4o | 31.0 | 34.0 |
| GPT-4o-mini | 17.0 | 22.3 |
| Qwen-2.5-72b-Instruct | 24.7 | 25.7 |
| Deepseek-V2.5 | 21.0 | 23.3 |
| Llama-3.1-70b-Instruct | 21.0 | 22.3 |

Figure 3: SWE-bench SWE-Search results

low rewards even when the Action Agent correctly identified the need for additional context, such as locating relevant files. This issue persisted despite the agent successfully identifying the files later. By implementing state-specific prompts across core state clusters (Searching, Planning, Editing), the value function became significantly more adept at interpreting the intent behind actions and evaluating their outcomes within each state. For further details on experiments distinguishing between effective and ineffective states, refer to **Appendix 8**.

**Scaling SWE agents with Inference-time Compute**   The success of large language models (LLMs) has traditionally been attributed to the expansion of training data and model size, i.e., training-time compute (Wei et al., 2022; Chung et al., 2022). Recently, researchers have started exploring how different methods scale with inference-time (OpenAI, 2024; Snell et al., 2024; Dubey et al., 2024). Here, we study the performance of software engineering agents through increased inference-time compute. As shown in **Figure 4b**, increasing search iterations leads to a consistent rise in the number of resolved issues. To ensure experimental feasibility across the 300 instances in the SWE-bench Lite dataset, we applied conservative parameters (maximum iterations = 100, maximum expansions per node = 3). Approaches like SWE-Search enable the allocation of greater resources to specific challenges, such as addressing critical software vulnerabilities (Rigaki et al., 2024; Fang et al., 2024), offering a scalable solution to complex tasks.

**Convergence of Value Function and Discriminator to Right Solution**   The search process can yield multiple proposed solutions. Ideally, the mean trajectory value of the the proposed solution that resolves the issue will always be the highest, which would yields the ideal performance of the agent (**Figure 3**). In practice, the value function successfully converged on the correct solution 73% of the time on average across the five models. The discriminator module performed even better, increasing the proportion of correct solutions selected to 84%. While in typical large action spaces, Monte Carlo Tree Search (MCTS) is run for thousands of iterations (Silver et al., 2016b),

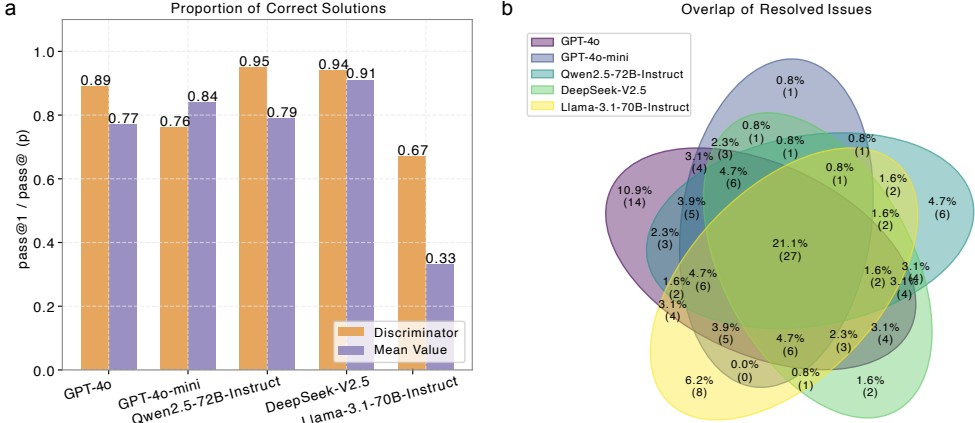

Figure 5: **(a) Value Function vs. Discriminator Comparison.** Comparison of value function vs. discriminator ability to discern the final solution that resolved the issue when there is one. The discriminator performs better across all models except GPT-4o-mini. DeepSeek-V2.5 had the smallest disparity between the two methods, suggesting an ability to act as a well-calibrated value function. **(b) Model-Specific Issue Resolution.** Venn diagram of resolved issues by model. Each model can solve a handful of unique instances.

the value function's success rate remains impressive given the computational constraints. However, SWE-Search could further benefit from enhanced methods for identifying the correct solutions more consistently, allowing it to fully reach its potential.

**Different Models can Resolve Vastly Different Issue Subsets**   When comparing the resolved instances across the five models, we observed significant diversity in the subsets of issues each model successfully solved. As shown in **Figure 5**, each model managed to resolve at least one unique instance. Notably, a surprising number of issues (33) were solved by other models but not by GPT-4o. This suggests that model diversity could play an important role, at least in the short term, in enhancing the performance of SWE-agents.

## 5    DISCUSSION AND CONCLUSION

In this paper, we introduced SWE-Search, a general framework that integrates Monte Carlo Tree Search (MCTS) and qualitative feedback to enhance the performance of software engineering agents. The proposed approach demonstrated improvements over different baseline models, highlighting the potential of search-based methods in software engineering tasks.

One of the key advantages of search-based approaches, as demonstrated in our work, is their ability to scale performance with increased inference-time compute. This flexibility allows the system to adapt to problems that require higher computational resources, such as discovering software vulnerabilities or even generating large codebases from scratch. Future research should focus on two main directions: (a) investigating how search agents scale with computational resources, and (b) expanding the application of software agent search to a broader range of complex use cases.

Given that search techniques like MCTS closely resemble the problem-solving processes of human software engineers, we expect these methods to become increasingly prevalent in agent-driven systems. As the nature of software engineering tasks evolves, system architectures will need to become more fluid and adaptable, fully leveraging the potential of search-based techniques. This evolution will likely lead to the development of larger, more general agentic systems capable of tackling a wide array of software engineering challenges.

## ACKNOWLEDGMENTS

Research was sponsored by the U.S. Army Research Office and accomplished under cooperative agreement W911NF-19-2-0026 for the Institute for Collaborative Biotechnologies.

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

## A    REPRODUCIBILITY

All models and data used in our work are publicly available. We additionally provide hyperparameter details in **Appendix 2**. The code will be released as a public repository upon publication.

## B    ADDITIONAL IMPLEMENTATION DETAILS

Moatless-Adapted is an extended version of the moatless-tools library with support for a tree structure, the ability to revert to earlier versions of the codebase, and the capability to run tests.

The standard implementation of moatless-tools is based on a finite state machine structure where a state holds information about file context and properties set in the configuration or from previous states. It can then transition to a new state when an action is executed. The request that initiates the action is created by an LLM. This follows a linear structure where one state can transition to another state. In Moatless-Adapted, this model is extended so that a state can expand by using actions to create more states. The connections between states are then represented in a tree structure with nodes.

Each state has a file context associated with it. This file context will be included in the prompt sent to an LLM. To limit the size of the prompt, files are divided into "spans," where a span could be, for example, a section of code (e.g., imports), a class, or a function. These are identified by span IDs. Thus, the LLM sees a limited part of the code at a time but can request more context by searching for or adding files and spans. The file context therefore changes over time, and a specific state of file context is linked to a specific state. In the standard implementation of moatless-tools, changes to the codebase are made linearly, and each change is saved directly to the file system. In Moatless-Adapted, however, there is a need to be able to revert to earlier states and thus return to a previous version of the codebase. To handle this, the code is stored in a git repository where each change is committed, and each state has a reference to a commit as well as the current patch of the diff from the initial commit that existed before starting. This way, one can go back to an earlier state by specifying the state ID, and the commit that was current at that time will be checked out.

The test files present in the file context are run each time the Plan state is initiated, and the test results are provided to the state. The tests are then run in Docker images built via the SWE-bench library. To use this approach in a benchmark where a larger number of instances should be able to run simultaneously, a solution is used where these images are run as pods in a Kubernetes cluster. Moatless-tools communicates with the testbed by applying patches and running commands via an API. When a new instance starts, a pod is created which is then reset at each run, applying the current patch and running tests according to the test command specified in the SWE-bench library. It's important to add here that the agent is not aware of the PASS_TO_PASS or FAIL_TO_PASS tests in the SWE-bench harness, but only knows how to run the tests. This corresponds to a real engineering environment where each project can have its own test commands.

## C    MCTS HYPERPARAMETERS

The Monte Carlo Tree Search (MCTS) algorithm used in this study employs several hyperparameters.

Table 2: MCTS Hyperparameters

| Hyperparameter | Description | Default |
|---|---|---|
| c_param | UCT exploration parameter | 1.41 |
| max_expansions | Max children per node | 5 |
| max_iterations | Max MCTS iterations | 100 |
| provide_feedback | Enable feedback | True |
| best_first | Use best-first strategy | True |
| value_function_temperature | Value function temperature | 0.2 |
| max_depth | Max tree depth | 20 |
| *UCT Score Calculation Parameters* | | |
| exploration_weight | UCT exploration weight | 1.0 |
| depth_weight | Depth penalty weight | 0.8 |
| depth_bonus_factor | Depth bonus factor | 200.0 |
| high_value_threshold | High-value node threshold | 55.0 |
| low_value_threshold | Low-value node threshold | 50.0 |
| very_high_value_threshold | Very high-value threshold | 75.0 |
| high_value_leaf_bonus_constant | High-value leaf bonus | 20.0 |
| high_value_bad_children_bonus_constant | High-value bad children bonus | 20.0 |
| high_value_child_penalty_constant | High-value child penalty | 5.0 |
| *Action Model Parameters* | | |
| action_model_temperature | Action model temperature | 0.2 |
| *Discriminator Parameters* | | |
| number_of_agents | Number of Discriminator Agents | 5 |
| number_of_round | Number of debate rounds | 3 |
| discriminator_temperature | Discriminator temperature | 1.0 |

These hyperparameters can be adjusted to fine-tune the MCTS algorithm's performance for specific problem domains or computational constraints. The values listed here are the defaults as defined in the `TreeSearchSettings` class and the MCTS implementation.

# D ABILITY OF MCTS TO ESCAPE UNPRODUCTIVE LOOPS VS. BASELINE

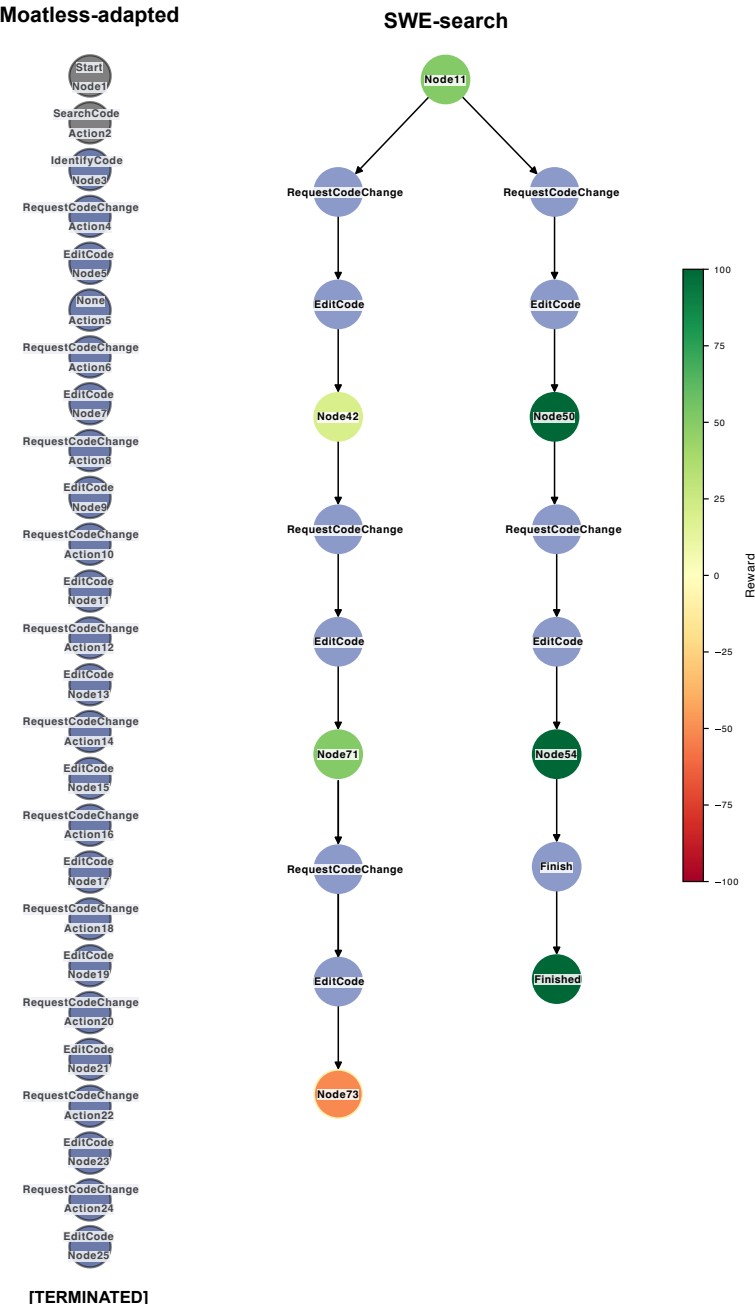

Figure 6: **Avoiding Repetitive Actions, django__django__10914.** We found that the base agent can often get stuck performing repetitive actions that do not bring it closer to solving the issue, and which commonly lead to unresolvable dead-ends. In this example, the base agent was stuck implementing wrong tests which continuously returned errors. In contrast, when this happens in SWE-Search, the Value Agent recognizes this, terminating these trajectories quickly, as happens in Node 73 (orange).

# E   MODEL INSTANCE RESOLUTION UNIQUENESS

To understand the complementary strengths of different models in resolving software issues, we analyzed how unique their resolved issue subsets where. Figure 7 illustrates the resolution patterns for each model across five of the codebases in SWE-bench-lite.

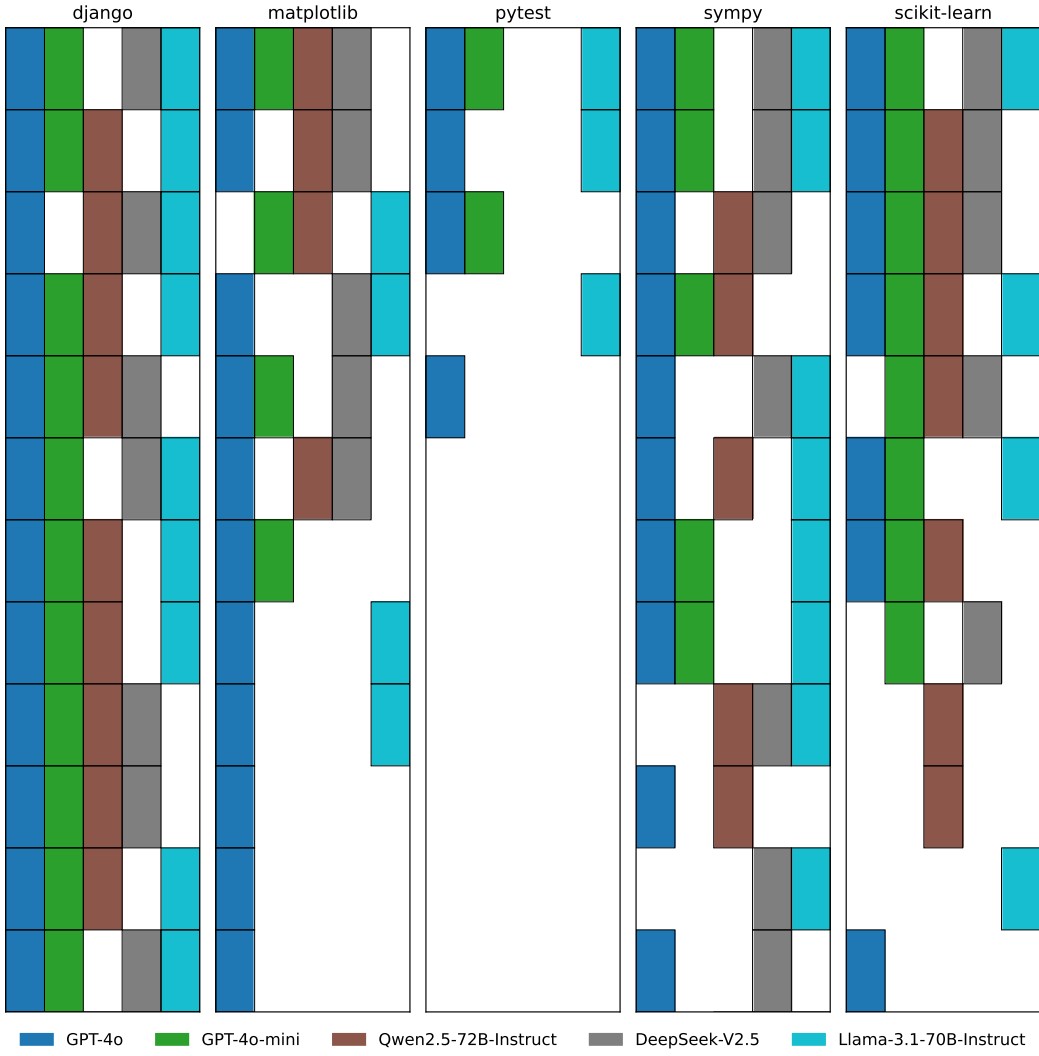

Figure 7: **Unique Issue Resolution Patterns Across Models and Libraries.** Each column represents a different Python repository, and each row within a column represents a specific issue. Colored blocks indicate successful resolution by the corresponding model (see legend). White spaces denote unresolved issues. This visualization highlights the diverse problem-solving capabilities of different models across various software domains, demonstrating that no single model dominates across all issues and libraries.

# F   ABILITY OF VALUE FUNCTION TO DISCERN SUCCESSFUL TRAJECTORIES

Before implementing SWE-Search, we conducted a general study across many models to evaluate the models' ability to differentiate states which led to resolved vs. unresolved issues. Figure 8 shows the results of this study. We found that in general, models assigned higher rewards to states which eventually led to resolved issues. Of particular interest was the Deepseek model, which seemed to identify critical errors in trajectories effectively. This was also observed in the final agent (see Fig. 5a).

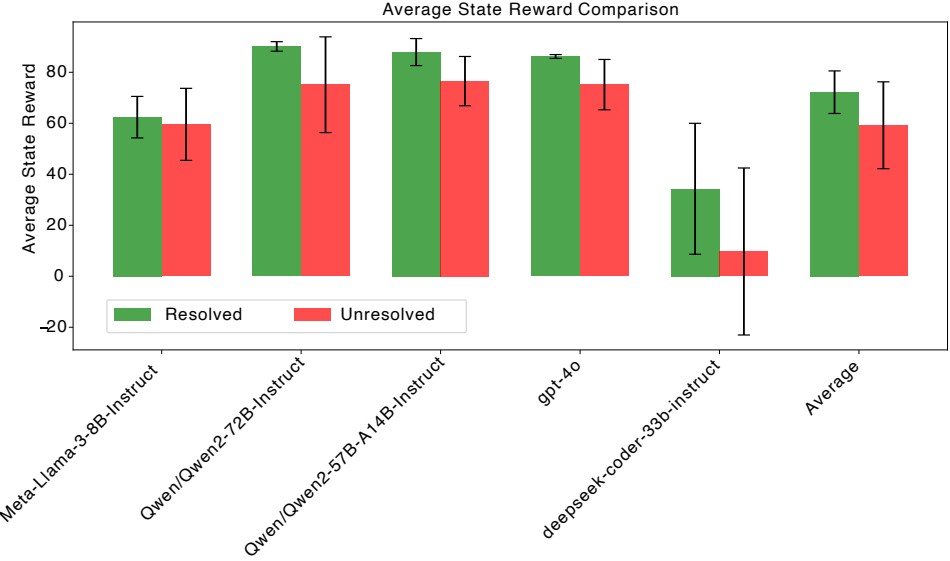

Figure 8: **Average State Reward Comparison Across Models.** This graph compares the average state rewards assigned by different language models for resolved (green) and unresolved (red) issues. Error bars indicate standard deviation. Most models consistently assign higher rewards to states leading to resolved issues, with the exception of the. The 'Average' column represents the mean across all models, demonstrating a clear distinction between resolved and unresolved states.

## G    VALUE FUNCTION PROMPTS

---

**Value Function Search Prompt**

Your task is to evaluate a search action executed by an AI agent, considering the search parameters, the resulting file context, and the identified code from the search results. Your evaluation will focus on whether the search action was well-constructed, whether the resulting file context is relevant and useful for solving the problem at hand, and whether the identified code is appropriate and helpful.

**You will be provided with four inputs:**

- **Problem Statement**: This will be provided within the `<problem_statement>` XML tag and contains the initial message or problem description the coding agent is trying to solve.
- **The Search Request**: This will be provided within the `<search_request>` XML tag and contains the search parameters used by the agent to define the search.
- **The Search Result**: The content retrieved based on the search parameters provided within a `<search_results>` XML tag.
- **Identified Code**: The specific code identified from the search results, provided within the `<identified_code>` XML tag.

**Search request parameters:**

- **File Pattern** (file_pattern): Glob patterns (e.g., \*\*/\*.py) to filter search results to specific files or directories.
- **Query** (query): A natural language query for semantic search.
- **Code Snippet** (code_snippet): Specific code snippets for exact matching.
- **Class Names** (class_names): Specific class names to include in the search.
- **Function Names** (function_names): Specific function names to include in the search.

**Evaluation Criteria:**

**Search Parameters:**

- Are they appropriately defined to focus the search on relevant files or code?
- Do they align well with the problem statement?

**Resulting File Context:**

- Does it contain relevant and useful information for solving the problem?
- Are there missing or irrelevant results indicating a need to refine the search?

**Identified Code Review (most crucial):**

- Is the identified code directly related to the problem?
- Does it provide the necessary functionality to address the issue?

---

**Overall Relevance:**

- Does the combination of search parameters, file context, and identified code effectively address the problem?
- Could there be a better approach or improvements?

**Reward Scale and Guidelines:**

Assign a single integer value between -100 and 100 based on how well the search action, resulting file context, and identified code addressed the task at hand. Use the following scale:

**100:**

- Search Parameters: Precisely match the problem needs; no irrelevant or missing elements.
- Identified Code: Completely and accurately solves the problem with no issues.

**75 to 99:**

- Search Parameters: Well-defined and mostly relevant; minor improvements possible.
- Identified Code: Effectively addresses the problem with minor issues that are easily fixable.

**0 to 74:**

- Search Parameters: Partially relevant; noticeable inaccuracies or omissions.
- Identified Code: Partially solves the problem but has significant gaps or errors.

**-1 to -49:**

- Search Parameters: Misaligned with the problem; poorly defined.
- Identified Code: Fails to address the problem effectively; may cause confusion.

**-50 to -100:**

- Search Parameters: Irrelevant or incorrect; hinders problem-solving.
- Identified Code: Unrelated to the problem; provides no useful information.

**Output Format:**

Please ensure your output strictly adheres to the following structure:
`<Explanation>` [A brief explanation of the evaluation in max one paragraph.]
`<Reward>` [A single integer reward value between -100 and 100]

---

**Value Function Plan Prompt**

Your role is to evaluate the executed action of the search tree that our AI agents are traversing, to help us determine the best trajectory to solve a programming issue. The agent is responsible for identifying and modifying the correct file(s) in response to the problem statement.

**Input Data Format:**

- **Problem Statement**: This will be provided within the `<problem_statement>` XML tag and contains the initial message or problem description the coding agent is trying to solve.
- **File Context**: The relevant code context will be provided within the `<file_context>` XML tag and pertains to the state the agent is operating on.
- **History**: The sequence of state transitions and actions taken prior to the current state will be contained within the `<history>` XML tag. This will include information on the parts of the codebase that were changed, the resulting diff, test results, and any reasoning or planned steps.
- **Executed Action**: The last executed action of the coding agent will be provided within the `<executed_action>` XML tag, this includes the proposed changes and the resulting diff of the change.
- **Full Git Diff**: The full Git diff up to the current state will be provided within the `<full_git_diff>` XML tag. This shows all changes made from the initial state to the current one and should be considered in your evaluation to ensure the modifications align with the overall solution.
- **Test Results**: The results of any test cases run on the modified code will be provided within the `<test_results>` XML tag. This will include information about passed, failed, or skipped tests, which should be carefully evaluated to confirm the correctness of the changes.

**Evaluation Criteria:**

**Code Correctness:** Evaluate whether the implemented code correctly addresses the problem. This includes verifying that the correct lines or sections of code have been identified and modified appropriately. Ensure that the changes are both syntactically and logically correct, and that the diffs accurately represent the intended modifications without introducing unrelated changes. Assess whether the modifications effectively solve the problem without introducing new issues or inefficiencies.

**Mistakes in Editing Code:** Identify any errors made during the code editing process. This involves checking for unintended deletions, incorrect modifications, or syntax errors introduced through the changes. Ensure that the Git diffs maintain integrity by only including the intended modifications and no accidental alterations to unrelated parts of the codebase.

**Testing:** Assess the proposed changes against existing test cases. Determine if the changes pass all relevant tests and evaluate whether any test failures could have been reasonably foreseen and avoided by the agent. Consider whether the agent anticipated potential test outcomes and addressed them proactively in the solution.

---

**History and Action Evaluation:** Review the agent's previous state transitions and actions

to determine if the current action contributes positively to solving the problem. Pay special attention to detect if the agent is engaging in repetitive actions without making meaningful progress. Evaluate whether the last executed action is appropriate and logical given the current progress and history of actions.

**Reward Scale and Guidelines:**

The reward value must be based on how confident you are that the agent's solution is the most optimal one possible with no unresolved issues or pending tasks. The scale ranges from -100 to 100, where:

**100:** You are fully confident that the proposed solution is the most optimal possible, has

been thoroughly tested, and requires no further changes.

**75-99:** The approach is likely the best one possible, but there are minor issues or opportu-

nities for optimization. All major functionality is correct, but some small improvements or additional testing may be needed. There might be some edge cases that are not covered.

**0-74:** The solution has been partially implemented or is incomplete or there are likely alter-

native approaches that might be better, i.e., this is likely not the most optimal approach. The core problem might be addressed, but there are significant issues with tests, logical flow, or side effects that need attention. There are likely alternative approaches that are much better.

**0:** The solution is not yet functional or is missing key elements. The agent's assertion that

the task is finished is incorrect, and substantial work is still required to fully resolve the issue. Modifying the wrong code, unintentionally removing or altering existing code, introducing syntax errors, or producing incorrect diffs fall into this range.

**-1 to -49:** The proposed solution introduces new issues or regresses existing functionality,

but some elements of the solution show potential or may be salvageable. Repetitive actions without progress fall into this range.

**-50 to -100:** The solution is entirely incorrect, causing significant new problems, or fails

to address the original issue entirely. Immediate and comprehensive changes are necessary. Persistent repetitive actions without progress should be heavily penalized.

**Output Format:**

Please ensure your output strictly adheres to the following structure:
`<Explanation>` [Your brief explanation of the evaluation in max one paragraph.]
`<Reward>` [A single integer reward value between -100 and 100]

---

**Value Function Request More Context Prompt**

Your role is to evaluate the executed action of the search tree that our AI agents are traversing, specifically for the RequestMoreContext action. This action is used when the agent requests to see code that is not in the current context, potentially revealing an understanding that relevant code is wholly or partially not visible, and enabling the agent to uncover important missing information.

**Evaluation Criteria:**

- **Relevance**: Are the requested files and code spans likely to be relevant to the problem at hand?
- **Necessity**: Is the additional context truly needed, or is the agent unnecessarily expanding the scope?
- **Specificity**: Has the agent been specific in its request, or is it asking for overly broad sections of code?
- **Contextual Understanding**: Does the request demonstrate a good understanding of the codebase structure and the problem domain?
- **Efficiency**: Is the agent making targeted requests, or is it asking for too much unnecessary information?
- **Progress**: Does this request seem likely to move the problem-solving process forward?

**Input Data Format:**

- **Problem Statement**: Provided within the `<problem_statement>` XML tag, containing the initial problem description.
- **File Context**: The current code context within the `<file_context>` XML tag.
- **History**: Previous state transitions and actions within the `<history>` XML tag.
- **Executed Action**: The RequestMoreContext action details within the `<executed_action>` XML tag, including the files and code spans requested.

**Reward Scale and Guidelines:** Assign a single integer value between -100 and 100 based on how well the RequestMoreContext action addresses the task at hand:

**100:** Perfect request that is highly likely to provide crucial missing information.

**75-99:** Good request with minor improvements possible in specificity or relevance.

**0-74:** Partially relevant request, but with noticeable inaccuracies or potential for better targeting.

**-1 to -49:** Poor request that is likely to provide mostly irrelevant information or expand the scope unnecessarily.

**-50 to -100:** Very poor request that is entirely irrelevant or demonstrates a fundamental

---

misunderstanding of the problem or codebase structure.

**Output Format:** Please ensure your output strictly adheres to the following structure:

`<Explanation>` [Your explanation of the evaluation in max two paragraphs.]
`<Reward>` [A single integer reward value between -100 and 100]

---

**Value Function Edit Prompt**

Your role is to evaluate the executed action of the search tree that our AI agents are traversing, with the goal of ensuring that a complete and verified solution is in place. The agent believes that it has finished solving the programming issue.

**Evaluation Criteria**

**Solution Correctness and Quality:** Verify that the proposed changes logically address the

problem statement. Ensure the changes fit contextually within the existing codebase without introducing new issues. Confirm syntactic correctness and that there are no syntax errors or typos. Assess whether the solution represents an overall improvement and is the most optimal approach possible.

**Accuracy of Code Modifications:** Check that the agent correctly identified the appropriate

code spans to modify. Ensure the changes made are accurate and do not include unintended modifications. Look for any alterations to unrelated parts of the code that could introduce new problems.

**Testing and Test Results Analysis:**

- **Importance of Test Updates:** It is crucial that the agent updated existing tests or added new tests to verify the solution. Failure to do so should be heavily penalized. The agent should ensure that code changes are validated by appropriate tests to confirm correctness and prevent regressions.
- **Assess Test Coverage:** Evaluate whether the agent has adequately tested the solution, including adding new tests for new functionality or changes. Verify that the tests cover relevant cases and edge conditions.
- **Penalization for Lack of Testing:** When calculating the reward, heavily penalize the agent if they failed to update or add necessary tests to verify the solution.

**Consideration of Alternative Approaches:** Always assess whether there could be a better

alternative approach to the problem. Mention any potential alternative solutions in your explanation if they are applicable.

**Identification and Explanation of Mistakes:** If the agent made incorrect actions, identify

exactly where and why the mistakes occurred. Explain the impact of any syntax errors, incorrect code modifications, or unintended changes.

**Assessment of Agent's Completion Assertion:** Verify if the agent's assertion that the task is finished is accurate. Determine if substantial work is still required to fully resolve the issue and address this in your evaluation.

**Input Data Format:**

- **Problem Statement**: This will be provided within the `<problem_statement>` XML tag and contains the initial message or problem description the coding agent is trying to solve.
- **File Context**: The relevant code context will be provided within the `<file_context>` XML tag and pertains to the state the agent is operating on.
- **History**: The sequence of state transitions and actions taken prior to the current state will be contained within the `<history>` XML tag. This will include information on the parts of the codebase that were changed, the resulting diff, test results, and any reasoning or planned steps.
- **Reasoning for Completion**: The reasoning provided by the agent for why the task is finished will be provided within the `<reasoning_for_completion>` XML tag. This includes the agent's explanation of why no further changes or actions are necessary.
- **Full Git Diff**: The full Git diff up to the current state will be provided within the `<full_git_diff>` XML tag. This shows all changes made from the initial state to the current one and should be considered in your evaluation to ensure the modifications align with the overall solution.
- **Test Results**: The results of any test cases run on the modified code will be provided within the `<test_results>` XML tag. This will include information about passed, failed, or skipped tests, which should be carefully evaluated to confirm the correctness of the changes.

**Reward Scale and Guidelines:**

The reward value must be based on how confident you are that the agent's solution is the most optimal one possible with no unresolved issues or pending tasks. It is important that the agent updated or added new tests to verify the solution; failure to do so should be heavily penalized. The scale ranges from -100 to 100, where:

**100:** You are fully confident that the proposed solution is the most optimal possible, has been thoroughly tested (including updated or new tests), and requires no further changes.

**75-99:** The approach is likely the best one possible, but there are minor issues or opportunities for optimization. All major functionality is correct, but some small improvements or additional testing may be needed. There might be some edge cases that are not covered.

**0-74:** The solution has been partially implemented or is incomplete, or there are likely alternative approaches that might be better. The core problem might be addressed, but there are significant issues with tests (especially if the agent did not update or add new tests), logical flow, or side effects that need attention.

**0:** The solution is not yet functional or is missing key elements. The agent's assertion that

> the task is finished is incorrect, and substantial work is still required to fully resolve the issue.
>
> **-1 to -49:** The proposed solution introduces new issues or regresses existing functionality,
>
> but some elements show potential or may be salvageable. Modifying the wrong code, unintentionally removing or altering existing code, introducing syntax errors, producing incorrect diffs, or failing to update or add necessary tests fall into this range.
>
> **-50 to -100:** The solution is entirely incorrect, causing significant new problems or failing
>
> to address the original issue entirely. Immediate and comprehensive changes are necessary. Persistent repetitive actions without progress, or failure to update or add tests when necessary, should be heavily penalized.
>
> **Output Format:** Please ensure your output strictly adheres to the following structure:
>
> `<Explanation>` [Your explanation of the evaluation in max two paragraphs.]
> `<Reward>` [A single integer reward value between -100 and 100]

## H    MOATLESS TOOLS STATE RIGIDITY

The Moatless-tools version (v0.0.2) enforces a rigid transition structure where agents must follow a specific sequence (search → identify → plan → edit). The implementation of this state transition system can be found here: [2].

### H.1    STATE TRANSITION SYSTEM

The transition system is configured through a function that accepts three optional parameters:

- `max_tokens_in_edit_prompt`: Controls the token limit for edit operations
- `global_params`: Defines parameters applicable across all states
- `state_params`: Specifies state-specific parameters

### H.2    STATE FLOW

The system defines a directed graph of states with specific transition rules:

1. **Search Phase** (`SearchCode`):
   - Initial state for code operations
   - Can transition to `IdentifyCode` upon successful search
   - Can move directly to `PlanToCode` when complete
2. **Identification Phase** (`IdentifyCode`):
   - Processes search results
   - Can return to `SearchCode` if needed
   - Progresses to `DecideRelevance` when finished
3. **Decision Phase** (`DecideRelevance`):
   - Evaluates identified information
   - Can trigger new searches

---

[2]`https://github.com/aorwall/moatless-tools/blob/8ec5d5193b6dce88ec6273c7ec31f9ea3a0bba6f/`
`moatless/transitions.py#L184`

- Transitions to planning when ready, excluding message field

This rigid structure ensures that tools are accessed in a predictable sequence, preventing conflicts while maintaining system integrity. Additional transitions defined in `CODE_TRANSITIONS` complete the state machine's behavior set.

## I    COST ANALYSIS

Table 3 presents the API costs for Moatless-Adapted and SWE-Search across different models. Search-based exploration of multiple solutions results in higher computational costs.

| Model | Moatless-Adapted | SWE-Search |
|---|---|---|
| GPT-4o | $40.86 | $576.00 |
| GPT-4o-mini | $9.90 | $52.34 |
| Qwen-2.5-72b-Instruct[*] | $8.50 | $42.50 |
| DeepseekCoderV2.5 | $3.66 | $18.37 |
| Llama-3.1-70b-Instruct[*] | $9.00 | $45.00 |

[*]Estimated costs based on comparable API pricing

Table 3: Cost comparison (USD) between Moatless-Adapted and SWE-Search

## J    COMPUTE-MATCHING ANALYSIS

Table 4 compares SWE-Search against compute-matched baselines. SWE-Search Pass@5 uses the 5 generated answers in 1 run, while for Moatless-Adapted uses the 5 generated solutions across 5 runs. We avoid doing the comparison on GPT-4o to avoid exorbitant API costs.

| Model | SWE-Search | | Moatless-Adapted |
|---|---|---|---|
| | Pass@1 | Pass@5 | Pass@5 |
| GPT-4o | 31.0 | 34.0 | - |
| GPT-4o-mini | 17.0 | 22.3 | 17.0 |
| Qwen-2.5-72b-Instruct | 24.7 | 25.7 | 22.3 |
| DeepseekCoderV2.5 | 21.0 | 23.3 | 22.0 |
| Llama-3.1-70b-Instruct | 17.7 | 22.3 | 21.7 |

Table 4: Performance comparison (%) between SWE-Search and compute-matched baselines

## K    INTERACTIVE DEMO

To help visualize the search process and provide transparency into our method, we provide an interactive demo at `https://streamlit.moatless.ai`. The demo presents a tree visualization where each node represents a state/action pair in the search process. Clicking on a node reveals detailed information including:

- Complete LLM interactions and tool calls
- State-specific value function outputs and reasoning
- Context information used for decision-making
- File changes and test results where applicable
- Test creation/execution and their outputs

This interface allows readers to explore how the search algorithm navigates through different states, makes decisions, and evaluates potential solutions. The visualization particularly highlights how state-specific value functions guide the exploration process and how the discriminator compares candidate solutions.

## L    DISCRIMINATOR DEBATE PROMPTS

---

**Solution Comparison Debate Round Prompt**

Below are a series of suggested changes to address the ¡Problem Statement¿. Your task is to carefully evaluate each change and decide which one is the most appropriate to address the issue. **Input Format:**

- **Problem Statement**: Original issue to be solved
- **Solutions**: Multiple candidate solutions, each with:
    - Unique solution ID
    - Git patch showing code changes

**Task:** Evaluate each proposed solution and identify which one most effectively addresses the

problem statement. Consider the completeness, correctness, and efficiency of each approach. **Output Format:**

- `<Explanation>`: Comprehensive explanation and reasoning behind your decision
- `<ID>`: The ID of the change you believe is the most appropriate

---

**Solution Comparison Conclusion Prompt**

Based on the Problem Statement and previous responses from the debate, determine the optimal solution. **Input Format:**

- **Initial Context**: Original problem statement
- **Agent Answers**: Evaluations and solution ID choices from the debate phase

**Task:** Synthesize the evaluations to identify the most appropriate solution. Report your

conclusion without referencing individual debate participants. **Output Format:**

- `<Explanation>`: Reasoning for the selected solution
- `<ID>`: Final chosen solution ID

---

