# OpenReview forum: "SWE-Search: Enhancing Software Agents with Monte Carlo Tree Search and Iterative Refinement"
_ICLR.cc/2025/Conference — ICLR 2025 Poster_

### Official Review · Reviewer_H9kG · 2024-11-01

**Soundness:** 2
**Presentation:** 2
**Contribution:** 2
**Rating:** 3
**Confidence:** 4

**Summary:**

This paper introduces SWE-search, a multi-agent framework that integrates Monte Carlo Tree Search (MCTS) with self-improvement mechanisms to enhance the performance of LLM-based software agents in dynamic repository-level tasks. The motivation is to  enable agents to iteratively refine their strategies using both numerical and qualitative feedback. By facilitating multi-agent debates, SWE-search optimizes the selection of solutions for complex problems through diverse perspectives. In experiments, the authors demonstrate the effeciveness of SWE-search by comparing it with a baseline called Moatless-adapted.

**Strengths:**

1. The motivation of this paper is clear, in the sense that software development is a non-linear process by its nature. The authors elaborate how their method mimic human work patterns in real world, which makes sense to me.

2. The idea of using MCTS to search for better solutions is novel.

**Weaknesses:**

1. The problem is not well-defined. Software development includes various kinds of tasks such as requirement analysis, architecture design, code generation from scratch, patching, debuging, testing, etc. The problem formulation part does not indicate which types of problems they are trying to sovle. In the experiment part, it seems that SWE-search aims to address issues in existing code bases. But it is still unclear how this problem is formulated as a sequential decision making problem. For example, in line 206, the authors describe the action space A_t as "a set of actions corresponding to type t". What EXACTLY are these actions? code generation? code repair? Overall, the problem formulation seems to be a general MDP-like formulation, with littel connections to software development tasks.

2. It is unclear how the value agent evaluate actions. It makes sense to me that the feedback is divided into an utility value and a piece of explanation. However, I doubt whether a normal LLM could provide accruate values to the SWE agent. Recall that in reinforcement learning, a value function is crucial to policy learning, and learning the value function is a very hard task. In software development, the value function should provide comprehensive testing results for given code and requirement list. Given the high-level descriptions in this paper, I am not convinced that a normal LLM could provide accurate feedback.

3. It is unclear how the MCTS runs. The authors only describe a way  to compute UCT in equation 3. However, many key information is missing, For example, what are the solutions we are searching for? What are the nodes (states) and edges (actions) in the context of software tasks? How to perform a node expansion?

4. Lack of justification of the Discriminator agent. Intuitively, if the value agent is accurate and the search algorithm works well, we would not need a multi-agent debate process.

5. Lack of baselines. Since the contributions of this paper mainly lie in the multi-agent framework, including the MCTS search, it is necessary to compare with existing Agent-level framework such as MetaGPT, ChatDev and GPT-Engineer. However, the authors only compare SWE-search with moatless-adapted.

**Questions:**

Please respond to the questions in the Weakness part.

---

> ### Author Response · Authors · 2024-11-22
>
> We'd like to thank you for reviewing our paper, and recognizing the novelty of our idea.
>
> **SWE-Agent, SWE-Bench, Baselines.** While we appreciate the reviewer's suggestions for additional comparisons, we note that tools like MetaGPT, ChatDev, and GPT-Engineer operate in fundamentally different settings from the SWE-bench benchmark [1,2]. Our paper builds specifically on repository-level software agents that can navigate and modify existing codebases - a distinct category from agents designed for code generation from scratch. Explaining in detail how software engineering agents work is beyond the scope of both our paper and this rebuttal. We have added Appendix H to point the reviewer to more implementation details of the base agent. We would be happy to clarify any specific aspects of the SWE-bench setting that would help contextualize our work.
>
> **Problem Definition & MCTS Operation.** The search tree operates over states representing repository and test execution context, with edges representing actions that progressively build toward a solution. The agent typically starts with Search actions to locate relevant files and functions, then uses Identify to examine specific code sections, before moving to Edit states for actual modifications. Each of these actions involves complex tool usage - for example, Search requires the LLM to output structured parameters like file patterns or queries for semantic vector search, while Edit involves generating code modifications and repository git diffs. The progressive nature of the task means errors in early stages (like searching the wrong files) can derail the entire solution process, making accurate value estimation particularly important. These tool calls are managed through a standardized interface that ensures the LLM outputs remain well-structured.
>
> **Value Function & State Prompts.** The effectiveness of our value function relies heavily on state-specific information and criteria, and providing the right context, such as file context, previous actions, and test outputs (Figure 1). As shown in Figure 4a, without specialized prompts, the value function fails to recognize the significance of key actions - assigning a low reward (0) to a successful code lookup action. With state-specific prompts, the same action receives an appropriate high reward (75), reflecting its importance in the solution process. This specialization allows the value function to achieve 73% accuracy in identifying correct solutions, with the discriminator further improving this to 84% (Figure 5a). The specialized prompts enable the value function to properly contextualize each state type (Search, Edit, Plan) and provide appropriate feedback aligned with the specific goals of that state. We have added the value prompts for each state in Appendix G.
>
> The discriminator process step complements traditional MCTS approaches. While our value function achieves 73% accuracy, direct comparison between candidate solutions improves that to 84% and offers unique advantages, and we think it’s one of the strengths of using LLMs as evaluators. This is particularly valuable when evaluating multiple promising solutions that may appear similar based on value estimates alone, or are not easily discernible in isolation.
>
> We are always available to answer any further questions that you may have.
>
> [1] J. Yang, C. E. Jimenez, A. Wettig, K. Lieret, S. Yao, K. Narasimhan, and O. Press, "SWE-agent: Agent-Computer Interfaces Enable Automated Software Engineering," arXiv:2405.15793, 2024.
>
> [2] C. E. Jimenez, J. Yang, A. Wettig, S. Yao, K. Pei, O. Press, and K. Narasimhan, "SWE-bench: Can Language Models Resolve Real-World GitHub Issues?," arXiv:2310.06770v3, 2024.

---

> > ### Author Response · Authors · 2024-11-24
> > **follow up on rebuttal response**
> >
> > Dear reviewer,
> >
> > We just want to gently remind you that the rebuttal deadline is Nov 26, which is tomorrow. We are wondering if our response has adequately addressed your concerns and would be happy to clarify or discuss any remaining questions you might have.
> >
> > Best, Authors

---

> > ### Comment · Reviewer_H9kG · 2024-11-27
> > **I have to maintain my score**
> >
> > Thanks for the rebuttal. However, I have to maintain my score as it is. The lack of sufficient technical details is still my major concern. Although the authors explained some of those in their responses, I do not believe the paper can be improved in a short period of time. Therefore, another round of reivew for this paper is necessary. Besides, the paper lacks of effecitve baselines, as is also pointed out by other reviewers.

---

> > > ### Author Response · Authors · 2024-12-03
> > >
> > > Thanks for your review.
> > >
> > > Unlike the other reviewers, we feel your feedback has been quite vague. We have provided ample technical details in our rebuttal, which you seem to not have recognized at all in your review. Furthermore, you have provided a confidence score of 4 while suggesting we compare with completely irrelevant benchmarks.
> > >
> > > We hope you can take this advice to improve yourself as a reviewer in the future. Thanks again for your time.

---

> > > > ### Comment · Reviewer_H9kG · 2025-04-21
> > > > **Congratulations**
> > > >
> > > > I thought this paper is clearly bellow the threshold so I did not follow up. Until recently I read it from social media that a 5533 paper got accepted at ICLR, which reminds me of this work.
> > > >
> > > > Frankly, I did not spend much time on this paper. Although I tried to figure out technical details in this paper, but the original version is like a messy coursework report. Why should I waste time on an unprepared submission? The purpose of a rebuttal is to address issues that reviewers may have misunderstood, rather than to guide you on how to improve your work. So I really don't need your 'advice' to teach me how to be a good reviewer. I will be when a submission deserves.
> > > >
> > > > About the baselines, I know existing works such as MetaGPT do not directly apply to SWE-Bench, but they can somehow be adapted to solve SWE-Bench by changing some roles and prompts, such as what is done in https://docs.deepwisdom.ai/main/en/blog/swebench/MetaGPT%20X%20Technical%20Report.html. Although many reviewers pointed out the issue about lacking of baselines, you still ignore.
> > > >
> > > > Since you have made it personal, I have no interest in discussing this paper any further. Again, congratulations on having a (possibly lowest rating ever) paper accepted to ICLR 2025!

---

### Official Review · Reviewer_rRDq · 2024-11-04

**Soundness:** 3
**Presentation:** 2
**Contribution:** 3
**Rating:** 5
**Confidence:** 4

**Summary:**

The paper proposes the use of MCTS and LLM self-evaluation to improve the performance of software agents. The agent achieves a strong 23% relative improvement on top of moatless-tools across the SWE-bench lite benchmark with a variety of different foundation models, and shows good results for test-time compute scaling with more environment steps.

**Strengths:**

- Very well-motivated and natural extension to coding agents
- Clear theoretical presentation of the RL/MCTS setting used
- Strong improvements across a wide variety of foundation models on the SWE Bench lite benchmark
- Promising results for scaling inference time compute with more environment steps

**Weaknesses:**

- Line 14: Unclear that the best software agents have “rigid processes”. Indeed, the meatless-tools framework that the authors use has a flexible choice of tools at every single timestep already. I agree with the author's justification that some search is necessary, but this claim should be more accurate.
- Completely unclear what value or reward function that value agent is optimizing in Section 3.1.2. I cannot find a definition for the value in the main paper. Appendix G is also unclear, hard to see why the value function would return values like 40, or 75 in the main paper in Figure 4. There needs to be far more analysis/ablations and justification of this design choice.
- Table 1 has no error bars or confidence intervals, which makes it hard to judge statistical significance.
- Experimental evaluation is limited to the lite dataset with a single agent (moatless-tools), it would strengthen the paper to show improvements on the full benchmark as well with other base SWE agents
- Unclear what prompts are used for the discriminator agent at the end, in general, the paper does not have enough details for full reproducibility
- Comparison to other SWE agents is missing.

Minor:
- The presentation is overly abstract and general until far later in the paper. It would be helpful for the authors to ground their initial theoretical presentation with concrete SWE examples from the start.
- Inconsistent use of \citep and \cite in the paper, please move in-text cites to \citep

**Questions:**

- How specific to the moatless-tools agent is the author’s algorithm? While the results on SWE-Bench are impressive, since then other agents have achieved 43% performance on the lite benchmark. (I recognize these agents may have been submitted after the author's results, but some comment on generality is needed)
- A natural baseline for scaling inference time compute would be pass@n and compute matching. Does the MCTS agent beat this simple baseline?

---

> ### Author Response · Authors · 2024-11-22
>
> Thank you for your thoughtful review.
>
> **Compute Matching Baseline.** We agree with the reviewer that such a baseline makes sense. We compare Moatless-Adapted pass@5 performance to that of SWE-search, which equates to the same compute over the benchmark. We observe that SWE-Search maintains strong performance compared to simple pass@n approaches, with even most models’ pass@1 outperforming the equivalent pass@5 Moatless-Adapted. This suggests our approach makes efficient use of compute resources, achieving better results through structured exploration rather than pure brute-force. We avoid running the GPT-4o benchmark due to the exorbitant cost.
>
> | Model | SWE-Search |  | Moatless-adapted |
> |-------|------------|------------|------------------|
> | | Pass @ 1 | Pass @ 5 | Pass @ 5 |
> | GPT-4o | 31 | 34 | - |
> | GPT-4o-mini | 17 | 22.30 | 17.00 |
> | Qwen-2.5-72b-Instruct | 24.70 | 25.7 | 22.3 |
> | DeepseekCoderV2 | 21 | 23.3 | 22 |
> | Llama-3.1-70b-Instruct | 17.70 | 22.3 | 21.67 |
>
> **Moatless-tools transitions.** We refrained from providing a detailed explanation of the base agent, as that would require a separate paper by itself, but have provided more details in Appendix B. The Moatless-tools version we’re comparing with (0.0.2) doesn’t have access to all the tools at any given time. Instead it has to follow a rigid transition structure (search -> identify -> plan -> edit), please see Appendix H for the exact lines of code that implement this.
>
> **Value Function.** The value function is optimizing what we refer to in the paper as the “expected utility” of each state/action. For each state type (search, identify, edit) what constitutes a good state/action is unique, and we therefore formulate specialized prompts for each state type that provide the relevant criteria, as well as detailed grading criteria for the values themselves. Appendix G aims to give a digestible overview of the structure of the prompts, but following the reviewer’s request we are adding the full system prompts for each state in the same section. As highlighted in the paper, this is one of the key insights to making this method work (Fig. 4a).
>
> The value function does indeed perform well in terms of separating states that lead to successful vs. unsuccessful trajectories (Appendix F, Fig. 8), as also indicated by the performance of the method including outperforming pure brute-force approaches. Having said that, we agree that beyond the empirical and analytical results we share, understanding the self-evaluation capabilities of swe-agents in more detail warrants further exploration that is befitting of an entire paper by itself.
>
> **Benchmarks / Generalizability.** SWE-Bench-lite is the official benchmark recommended by the SWE-Bench team.
>
> In terms of transferability, our algorithm is indeed easily transferable to other OS agents, as none of them currently utilize search. Furthermore, the fluid nature of Moatless-Adapted agent framework we base SWE-Search on in its current state can be easily extended to incorporate tools and functionalities from other agents. One of the main novelties of the agent is the ability to build a git commit tree of repositories, in order to flexibly transition between any of them, which was a notable engineering challenge. We have attached an anonymized version of the open-sourced repository.
>
> More benchmarks are always desirable. Still, focussing on different agent implementations would take focus away from the algorithm we propose. We contend that the consistent increase on the  moatless-tools baseline is sufficient evidence of the potential of SWE-Search, and we think the OS implementation and lessons learned will be a valuable contribution to the community.

---

> > ### Comment · Reviewer_rRDq · 2024-11-23
> >
> > Thank you for your response. I am not satisfied that your response has addressed the concerns in the review. For example:
> >
> > > Table 1 has no error bars or confidence intervals, which makes it hard to judge statistical significance.
> >
> > Has no response.
> >
> > > Unclear what prompts are used for the discriminator agent at the end, in general, the paper does not have enough details for full reproducibility
> >
> > Has no response.
> >
> > > Completely unclear what value or reward function that value agent is optimizing in Section 3.1.2.
> >
> > I see you have alluded to this but provided no further detail.
> >
> > As things stand, my original concerns remain and I am unable to change my rating.

---

> > > ### Author Response · Authors · 2024-11-24
> > >
> > > Thank you for engaging in discussion with us.
> > >
> > > We'd first like to highlight the key contribution of SWE-Search: Providing a method and implementation to integrate in standard available SWE-Agents with tree search. We employ a modified Monte-Carlo algorithm for agents, similar to agentic search approaches primarily applied to web agents [1]. As other reviewers note, novelty arises in the way these ingredients (SWE-Agents, MCTS) are *combined* and this is the primary focus of our work.
> > >
> > > The variance across multiple runs is not a primary concern in any agentic benchmarks and is typically not reported in MCTS agent implementations [1], web agents [2] (ICLR 2024 Oral), or SWE-agents/SWE-bench [3]. Therefore we believe this should not form grounds to reject our work, especially given the below evidence which reinforces the significance of our method.
> > >
> > > **Statistical Significance.** There is ample evidence of the statistical significance of our approach. The method shows strong consistency across 5 models (GPT-4o, GPT-4o-mini, Llama, Qwen, DeepseekCoder) with a moderate coefficient of variation (0.23) and substantial double-digit gains (13.0-24.3%). This consistency, combined with SWE-Search consistently outperforming compute-matched baselines (pass@5), is strong evidence for the method's effectiveness. Given prior works, and the evidence above, multiple search runs for each model would significantly increase the cost of our study without providing any significant additional value.
> > >
> > > **Discriminator Prompts.** We use the exact format for multi-agent debate used in the previous works alluded to in the paper [4]. We have now added the detailed prompt template in **Appendix L** and point to it in the paper.
> > >
> > > **Value Function.** Following the reviewer's initial comments, we have already provided a detailed and extensive Appendix G section for the value function, which contains the exact criteria that the value function is optimizing for at each state. The value function is given as:
> > >
> > > $$ V(s_t, a_t, s_{0:t-1}, a_{0:t-1}) $$
> > >
> > > where $s_t$ is the current state, $a_t$ is the current action, and $s_{0:t-1}, a_{0:t-1}$ represent the history of states and actions. This function optimizes a composite reward (-100 to +100) based on state-specific evaluation criteria detailed in Appendix G. For Search states, this includes query quality and context relevance; for Edit states, code correctness and test coverage. The function produces both a numerical value and an explanation to guide exploration, achieving 73% accuracy in identifying successful trajectories (Section 4.2).
> > >
> > > Thank you once again for reviewing our paper and engaging with us. If there remain any additional points you'd like to bring up, we are happy to address them.
> > >
> > > [1] Zhang, Y., et al. (2024). WebPilot: A Versatile and Autonomous Multi-Agent System for Web Task Execution with Strategic Exploration. arXiv preprint arXiv:2408.15978.
> > >
> > > [2] Gur, I., et al. (2024). A Real-World WebAgent with Planning, Long Context Understanding, and Program Synthesis. In International Conference on Learning Representations (ICLR) 2024 (Oral).
> > >
> > > [3] Zhang, Y., et al. (2024). AutoCodeRover: Autonomous Program Improvement. In Proceedings of the International Symposium on Software Testing and Analysis (ISSTA) 2024.
> > >
> > > [4] Du, Y., et al. (2023). Improving Factuality and Reasoning in Language Models through Multiagent Debate. arXiv preprint arXiv:2305.14325.

---

> > > > ### Comment · Reviewer_rRDq · 2024-12-03
> > > >
> > > > I thank the authors for responding and providing more clarifications to their algorithm. These details are essential and without them, the proposed algorithm is completely not reproducible. E.g. the details in Appendix G are vital to the algorithm and only now included.
> > > >
> > > > > The variance across multiple runs is not a primary concern in any agentic benchmarks and is typically not reported in MCTS agent implementations
> > > >
> > > > A truly terrible trend given that LLM sampling is stochastic. This goes against scientific practice and was in fact the focus of a recent critique of the field [1]. I hope to see future work in this field go against this worrying trend.
> > > >
> > > > I have read the other reviewer's concerns and believe my current score is fair. I wish the authors good luck.
> > > >
> > > > [1] Adding Error Bars to Evals: A Statistical Approach to Language Model Evaluations. Evan Miller.

---

> > > > > ### Author Response · Authors · 2024-12-03
> > > > >
> > > > > You could be right. It's just to expensive to run multiple times over the whole benchmark, so SWE-bench results only require one run. Even a difference of 1-2% is pretty significant on this benchmark though. We will work hard to provide more thorough results to alleviate such concerns. Thank you again for your constructive comments.

---

### Official Review · Reviewer_TtHv · 2024-11-04

**Soundness:** 2
**Presentation:** 1
**Contribution:** 3
**Rating:** 5
**Confidence:** 3

**Summary:**

The authors propose a new agentic system for software development applications based on frontier models and MCTS. They show that this leads to a performance improvement relative to a separate baseline model across several different LLM engines.

**Strengths:**

- The topic is very relevant and timely.
- An improvement on the problem tackled here has incredible potential.
- MCTS is a good approach to take for this kind of problem, and finding a good way to integrate it has been thought about a lot, so this work is quite useful to the community.
- This approach is modular enough that it could feasibly be integrated effectively with other approaches, leading to rich future work.

**Weaknesses:**

- I think the Brown M&M here is really the citation used for MCTS. MCTS was absolutely not proposed for the first time in Silver et al.'s seminal work (Silver even cites the original paper), and using that here hints that this work is very unpolished. But this citation is just one of the symptoms of a greater problem: this paper is poorly written to the point that it reads more like an advertisement than a technical, scientific paper (lines 346-348 are a good example, but the issues are throughout and I'm not confident this can be solved without a full rewrite). While clarity is often overlooked, I'm not sure the rest of the paper is of the exceptional standards needed to justify getting away with it.
- There are serious typos in a few places (e.g., the paragraph at 92 or all the missing parentheses around citations). I think the work needs---at a bare minimum---a full run-through to correct all these.
- The primary area listed here is wrong. The taxonomy here refers to autonomy and planning in the robotics domain. This work has nothing to do with that.
- The paper only mentions its tested on SWE-Bench Lite later on. That should be clarified in the abstract or, at least, in the introduction. That seems like a big difference to me here.
- There's only a single baseline here (Moatless). I would have expected, say, three different baselines. The multiple LLMs here serve the role of something similar to random seeds in demonstrating that the results aren't a statistical fluke.
- The performance improvement with the baseline seems quite incremental despite all the additional complexity.
- "general value functions" are a particular concept in RL research. The authors need to change that term out.
- SWE-Bench really is not a comprehensive benchmark and should not be said to be one. Not that it's a bad benchmark here, but you should remove the statement saying it is.
- It would be good to swap out the lines in Figure 4(b) and the bars in 5(a) with different styles to make the paper more friendly to both colorblind readers and black and white printouts.

The above is my reading of this work, and unfortunately, I'm not really sure how much ground the authors can make with me if they try to argue against it (do please still try, though, as there's no point not doing so). However, the reviewing process is noisy, and no matter how anyone feels about their evaluation, the data says that even the best reviewers can only provide truly medium-confidence reviews. Thus, I'm inclined to acquiesce and change my score if the other reviews disagree.

**Questions:**

Please respond to the above when possible as well.

- What is the exact data flow here? There are a few modules, but how do they actually connect together? The text is quite sparse on that and seems to instead focus on describing only the virtues of their approach.
- What's the change in cost here? How much more expensive is this approach compared to other ones? I would like to see something like Table 1 but with USD API costs.
- Is that doing significantly better on SWE-Bench than the norm? A quick look says no (but they look like contemporary works, so I won't hold it against you, and it is optional to address this question). Would this naturally integrate with some of those approaches?
- Line 479-481 seems highly significant at first glance. Is there a reason that isn't a bigger topic in the paper?
- Can you give more details on the difference between your work and the Alibaba system? Is that contemporaneous work?

---

> ### Author Response · Authors · 2024-11-22
>
> We’d like to express our appreciation for such a thorough review of our paper.
>
> **Citation.** The citation was not intended to imply that MCTS was introduced by Sliver et. al, but rather that it was the first to combine it with DL approaches. Nevertheless, as big Van Halen fans, we acknowledge the reviewer’s analogy and update our citation accordingly.
>
> **Writing.** Additionally, we have done an extensive review of the paper, improving and simplifying the writing, rearranging the methods section to follow a consistent and more interpretable format, removing the term general value functions and addressing all points brought up by the reviewer.
>
> **How agent works/weaknesses.** Explaining how the whole agent (or any agent) implementation works in detail is very hard without diving into the codebase itself. We explain more about the core agent implementation in Appendix B. In general, the implementation is indeed quite fluid, with the agent itself deciding what to do next given definitions to a set of actions in its context. Explaining how the base agent works would need a paper of its own entirely, and we instead focus on explaining the algorithm we propose. Is there a specific point in the pipeline you’d like us to clarify further?
>
> **Limitations.** An example of a current limitation is demonstrated in Figure 4a. The value functions require carefully crafted state-specific prompts to perform effectively. Getting language models to serve as reliable value functions across diverse domains (e.g., coding, web navigation) currently requires manual adaptation of these prompts. Future work should focus on automating this process, potentially enabling more general and adaptable value estimation across different task domains.
>
> **Improvement.** We believe a 23% average increase across 5 models is quite significant for this benchmark (other reviewers seem to agree), nevertheless this is a matter of opinion. Virtually all the top solutions on SWE-bench were submitted after we wrote the paper, and utilize Anthropic’s new model. The moatless-tools base agent using this new model achieved a 38%, which is nearly on-par with SOTA (submitted to official benchmark). Our current results also achieve the best performance for open-source models through Qwen-72b. The highlight of our algorithm is that it's designed to be model and agent-agnostic. We intentionally kept the value function, node tree expansion, and search separate to the base agent, exactly in order to enable the algorithm to transfer to other agents (please see provided code). This took considerably more effort than hardcoding the algorithm within the base moatless tools agent.
>
> **Inference-time Scaling (479-481).** Those lines point to the direction we hope the community will expand works like ours in the future. Search approaches for code essentially enable the “scaling” of swe-agents according to how extensively you want to explore the space of available solutions. Additionally, tree search gives added flexibility to an Agent, allowing it to “fail” and simply backtrack and learn from experience, essentially making the agent more open-ended. While this requires additional compute and may be overkill in some scenarios, there are domains like cybersecurity or scientific analysis where exploration far outweighs efficiency concerns - similar to how MCTS is used in Chess and Go. Beyond the scaling results in Figure 4b, we leave deeper exploration of this direction to future work, as it requires specialized benchmarks and evaluation frameworks that don't yet exist. The codebase we have open-sourced provides a foundation for exploring these directions.
>
> **Alibaba Agent.** The Alibaba system uses MCTS to explore the codebase and provide potential locations that could be responsible for the issue. This information is then passed along to a code agent which generates a code patch. This second stage follows a linear structure, which is different from how our agent works, where it is allowed to flexibly expand and transition to any state/action at any point, including independent code edits, by utilizing a git commit tree.
>
> **API Costs.** You raise an important point about API costs. As expected, the search-based exploration of multiple solutions results in higher computational costs. We also note that costs can be significantly reduced by using open-source models like Qwen-72b, which achieve competitive performance at a fraction of the cost of closed-source alternatives. Additionally, as we show in the compute-matching baseline, matching SWE-Search outperforms a naive baseline using the same amount of compute by executing multiple runs.
>
> | Model      | Cost ||
> |------------|------------|------------|
> |            | Moatless-adapted | SWE-Search |
> | GPT-4o      | $40.86     | $576.00    |
> | GPT-4o-mini | $9.90      | $52.34     |
> | Qwen       | *$8.50*     |*$42.50*      |
> | DeepSeek   | $3.66      | $18.37     |
> | LLaMA      | *$9.00*      | *$45.00*     |

---

> > ### Author Response · Authors · 2024-11-24
> > **follow up on rebuttal response**
> >
> > Dear reviewer,
> >
> > We just want to gently remind you that the rebuttal deadline is Nov 26, which is tomorrow. We are wondering if our response has adequately addressed your concerns and would be happy to clarify or discuss any remaining questions you might have.
> >
> > Best, Authors

---

> > > ### Comment · Reviewer_TtHv · 2024-11-26
> > > **Paper Over Page Limit**
> > >
> > > I'll provide my full response in a bit, but are you aware that the paper is now 11 pages? While they've extended the discussion period, they haven't extended the time during which you're allowed to make edits. You need to correct this before this earlier deadline or my response and those of the other reviewers are meaningless.

---

> > > > ### Author Response · Authors · 2024-11-26
> > > >
> > > > My sincerest apologies for the oversight. We have fixed the page limit issue and re-uploaded the draft. Thanks for bringing this to our attention!

---

> > > > > ### Comment · Reviewer_TtHv · 2024-11-27
> > > > > **Better but Still Lacking**
> > > > >
> > > > > I wish I could be more positive about this paper because I really do think this direction has potential. However, I agree with the other reviewers that this:
> > > > >
> > > > > > There's only a single baseline here (Moatless). I would have expected, say, three different baselines. The multiple LLMs here serve the role of something similar to random seeds in demonstrating that the results aren't a statistical fluke.
> > > > >
> > > > > is a pretty big problem. Even just using the pure base models with minor edits (to get it above 0% scoring) as a baseline would have been better than having a single comparison point. It's hard to argue the method isn't incremental when there's only one comparison point.
> > > > >
> > > > > Likewise, this:
> > > > >
> > > > > > How agent works/weaknesses. Explaining how the whole agent (or any agent) implementation works in detail is very hard without diving into the codebase itself. We explain more about the core agent implementation in Appendix B. In general, the implementation is indeed quite fluid, with the agent itself deciding what to do next given definitions to a set of actions in its context. Explaining how the base agent works would need a paper of its own entirely, and we instead focus on explaining the algorithm we propose. Is there a specific point in the pipeline you'd like us to clarify further?
> > > > >
> > > > > is still an issue. I agree that agentic systems are not straightforward to visualize or explain cleanly. But it can be done and indeed must be done here. I work with agentic systems, and I'm an expert in RL, but I struggle to understand from Section 3 how the modules work and the nature of their interactions (even in the new version). If myself and the other reviewers can't clearly grasp the method from the main text, that's a pretty big problem. This really can't be relegated to the appendix.
> > > > >
> > > > > The current explanations in the main text are very fluffy with lots of repetition and lots of non-technical text (your paper is investigating a method and looking at its pros and cons, not trying to sell it to me) and could certainly be condensed with the free space used to draw a nice flowchart (which may need to be an oversimplification) with the text connecting to that.
> > > > >
> > > > > Also, this:
> > > > >
> > > > > > Citation. The citation was not intended to imply that MCTS was introduced by Sliver et. al, but rather that it was the first to combine it with DL approaches. Nevertheless, as big Van Halen fans, we acknowledge the reviewer's analogy and update our citation accordingly.
> > > > >
> > > > > does not seem to have been handled. In the paper, you don't cite this as a "Deep learning MCTS" but as "MCTS." In this way, you're only citing a secondary source here (Silver et al. 2016) and not the originals (see 11, 12 in Silver et al. 2016). In this form, your citation implies that Silver et al. 2016 invented MCTS---your intention isn't important here as the reader does not have access to . You must always cite the primary source and then any secondary sources you used to understand that primary source.
> > > > >
> > > > > I think you may have misunderstood the Brown M&M analogy. What that means here is that this MCTS citation being so wrong means I'm not altogether confident that the rest of the citations here (which may be in areas less well-known to me) are valid. Certainly, the related works section doesn't mention any consideration of computers writing computer programs prior to 2024. Am I to understand that nobody used any language models to write any kind of code prior to 2023?
> > > > >
> > > > > Don't misunderstand me, though. I appreciate the author's work, and I think the paper is a fair bit stronger than what was originally submitted. Right now, this is closer to what I expect to see at ICLR. But in its current form with all the issues above, I don't think this is ready yet, and a new suite of man experimental results alongside a large rewrite would take a month or more to accomplish. I'm raising my score slightly to reflect my changed opinion.

---

> > > > > > ### Author Response · Authors · 2024-11-30
> > > > > >
> > > > > > Thank you very much for your response.
> > > > > >
> > > > > > > There's only a single baseline here (Moatless). I would have expected, say, three different baselines. The multiple LLMs here serve the role of something similar to random seeds in demonstrating that the results aren't a statistical fluke.
> > > > > >
> > > > > > The question of seeds is still an issue for us because of how expensive it is and the little difference it will make to the results, and how this is not done on any SWE-bench papers.
> > > > > >
> > > > > > Same goes for benchmarks, we'll basically have to implement the algorithm with multiple agents, with multiple seeds to satisfy what the current reviewers are suggesting in order to do apple-to-apples comparison with the base agents or just plain achieve SOTA on the benchmark (aka SOTA chase instead of focus on the algorithm).
> > > > > >
> > > > > > > Am I to understand that nobody used any language models to write any kind of code prior to 2023?
> > > > > >
> > > > > > SWE-agents are an extremely new area of study, so yes, **repository-level** software agents are something that's been introduced in the last two years. We acknowledge that the tone of the writing could have used some refinement, which we have done using your feedback, but using the analogy to claim that our whole paper's credibility is put into question is a little hyperbolic.
> > > > > >
> > > > > >
> > > > > > Having said that, we appreciate you as the most responsive and thorough of the reviewers. We respect the time and effort in your review even though it did not yield the outcome we would have wanted or expected.

---

### Official Review · Reviewer_j8qB · 2024-11-04

**Soundness:** 2
**Presentation:** 2
**Contribution:** 3
**Rating:** 3
**Confidence:** 3

**Summary:**

Large language models have been used to automate repository-level tasks in software engineering. Current models used rigid, hand-crafted rules for decision making in these tasks. However, these constraints can cause these software agents to be less adaptable to long-horizon tasks. This paper proposes SWE-search, a framework incorporating multiple agents and Monte Carlo tree search (MCTS) to perform these software tasks. This framework consists of a search agent to perform exploration over software actions, a value agent for self-learning feedback and a discriminator agent for deciding on actions. The paper also includes empirical comparisons of the proposed approach against currently used software agents.

**Strengths:**

**Originality:** The three types of agents being trained for the framework are based on pre-existing models. However, novelty arises in the way they are combined to form a automated SE framework.

**Significance:** The use of LLMs for automating SE tasks is already pretty widespread. Significantly improving on the current SOTA would further increase their adoption into more use cases.

**Clarity:** The idea presented was clearly laid and explained well.

**Quality:** The problem is well-motivated and plainly laid out. The intuition behind the proposed framework is conceptually simple and the way the multiple agents in the framework are used is sound and reasonable. The experimental design was sound and the baselines against which their approach was compared were sensible. I appreciated the demonstration for the importance of state information in Sec 4.2.

**Weaknesses:**

There are however a few concerns that need to be addressed.

1. There were a number of (I think) formatting issues. This resulted in some the equations not making sense.
    - the arguments in the value function on line 193 do not make sense
    - the expected cumulative reward on line 196 also does not make sense.
    - $O$ is never defined

2. Section 3 was very confusing:
    - $t$ is used to refer several different things. In 3.1 it is a state and later it is used to define time step. In section 3.1.1, $t$ is used to refer to action type.
    - In 3.1.1, actions, states and action-types seem to be used interchangeably. I can't tell if the action-types are the states or actions. Fig 1 suggests they are the states in the search tree so why not call them that in all instances?
    - $T$ is not defined on line 190.

3. In Sec 4, I'm not exactly sure what the moatless-adapted baselines entails. I read after under the impression that they are the models using the moatless tools with the strict search restrictions.

4. There is no Table 3. There is, however, a Figure 3 that is a table that is never referenced but the values do not align with the descriptions of Table 3.

5. MCTS is stochastic algorithm. As such the evaluations should be performed using more than one random seed on the evaluation set. This would provide the expected performance wrt solve rate, computation, etc. Most importantly, it would provide some measure of significance testing to see if the results support the claims made by authors about the performance gains of SWE.

**Questions:**

1. Line 181: Typo in "including"

2. In Equation 1, why is value function not dependent on the context space?

3. In Equation 2, in search and RL, a balance between emphasizing the importance of long-term and short-term rewards is made using a discount factor. The $early\textunderscore depth\textunderscore bonus$ and $late\textunderscore depth\textunderscore penalty$ appear to be an attempt to handcraft this balance. Why not just use discounting?

4. Lines 411-415: I do not understand what that segment is trying to say.

---

> ### Author Response · Authors · 2024-11-22
>
> Thank you for your detailed review and acknowledging the novelty of our method and the potential significance it can have by advancing SWE-agent approaches. We'd like to offer the following information in response to your comments.
>
> **Writing.** We updated the expected cumulative reward to properly index the states and actions with respect to time, and correctly formatted the conditional expectation.
>
> The value function equation and notation has been revised for consistency throughout the paper. We now use $V(s_t, a_t)$ instead of $O_n(s_n, a_n)$ to represent the value function estimate of a state-action pair.
>
> Actions, which are separated into action types, cause transitions to State types. These states hold all the environment variables including state of repository, potential test outputs, model context, and previous actions in the trajectory. We make this distinction more clear in the manuscript (3.2), and update equations accordingly to make this more explicit.
>
> Changed Figure 3 -> Table 3. Figure 3 indeed demonstrates the “ideal performance” of the method. Since we conclude MCTS after 5 finished states, pass@5 is the performance of the agent if the correct solution was selected 100% of the time, when present.
>
> **Multiple runs.** We appreciate the suggestion regarding multiple random seeds, but maintain our evaluation methodology is appropriate given: (1) the magnitude of improvements (mean +23%) far exceeds what could be attributed to MCTS randomness, with consistent positive deltas from +17% to +27% across five different architectures over 300 benchmark instances. The compute-matching baselines reinforce this notion. (2) software engineering benchmarks (HumanEval, MBPP, SWE-bench) traditionally focus on solve rates rather than statistical significance [1] given the binary nature of problem-solving success, and so do search approaches for other agentic tasks [2] (3) running multiple seeds would multiply computation costs without meaningfully changing the practical implications.
>
>
> **Q2:** Since our state transitions $P : S × A × C → ∆(S)$ incorporate context information, and each state contains the relevant context from previous transitions, it would be redundant to include C explicitly in the value function. This follows standard formulations in POMDPs where the state representation is designed to be complete with respect to the information needed for decision-making.
>
> **Q3:** Traditional discounting (something of the form $γ^d$) wouldn’t be able to readily encourage the types of behaviors we were looking for. For the ‘early_depth_bonus’, we wanted to give a very high bonus that quickly vanishes, while for the ‘late_depth_penalty’ we wanted to penalize depth more slowly. Thus we found the approach of separating these two easier to work with.
>
> **Q4:** In the original agent, the transitions between state/action types are rigid. This provides some guardrails for the agent in order to continue along a reasonable path, concretely: *search* -> *plan* -> *code* -> *finish*. By allowing transitions between any state/action types, (for example transitioning from *plan* back to *search*) we are able to give more flexibility to the agent, but at the same time risk the agent making erroneous decisions like searching over and over again, making wrong transitions (instead of editing the right files, searching again and ending up with wrong files), or never concluding (reaching a finished state), for example by continuously introducing new edits or searches. Please see Appendix D to see an example of an agent getting stuck in these loops. We have also updated the paper to make this clearer (Section 3.2).
>
> We are always available to answer any further questions you may have.
>
>
> [1] J. Yang, C. E. Jimenez, A. Wettig, K. Lieret, S. Yao, K. Narasimhan, and O. Press, "SWE-agent: Agent-Computer Interfaces Enable Automated Software Engineering," arXiv:2405.15793, 2024.
>
> [2] Y. Zhang, Z. Ma, Y. Ma, Z. Han, Y. Wu, and V. Tresp, "WebPilot: A Versatile and Autonomous Multi-Agent System for Web Task Execution with Strategic Exploration," arXiv:2405.15793v3, 2024.

---

> > ### Author Response · Authors · 2024-11-24
> > **follow up on rebuttal response**
> >
> > Dear reviewer,
> >
> > We just want to gently remind you that the rebuttal deadline is Nov 26, which is tomorrow. We are wondering if our response has adequately addressed your concerns and would be happy to clarify or discuss any remaining questions you might have.
> >
> > Best, Authors

---

> > > ### Comment · Reviewer_j8qB · 2024-11-26
> > > **Acknowledgement of rebuttal**
> > >
> > > I appreciate the authors' response. However, I still maintain my score. Given that
> > >
> > > i) there is still only one competitive baseline,
> > > ii) a lack of multiple seed evaluations (I understand the attempt to add some significance by using different architectures but I remain unconvinced about statistical significance),
> > > iii) and the selection of the two search bonuses are just a pair of default values with no sweeps,
> > >
> > > the authors' claims that this is a general framework is weak. Really, all one can say at this point is that their approach provides an incremental improvement for the SWE-bench Lite task.

---

> > > > ### Author Response · Authors · 2024-11-30
> > > >
> > > > Thank you for acknowledging our responses and reviewing our paper. We will take your feedback into account to improve our work.

---

### Author Response · Authors · 2024-11-22
**Thank you for your time and constructive comments.**

Thank you to all the reviewers for their time and constructive comments. We are thankful to have received detailed feedback to improve our manuscript. Furthermore, we were happy to see all reviewers recognize the novelty and potential utility of our approach, particularly in introducing the first tree search method for repository-level software tasks.

The reviewers made many comments regarding the writing.  Following your feedback the manuscript has been extensively re-edited,  particularly from the abstract through the methods section, to address your concerns point-by-point. We address these revisions in our respective comments individually.

In response to reviewer requests, we have added four new sections: implementation details (Appendix H), cost analysis (Appendix I), compute-matched baselines with Moatless-Adapted (Appendix J), and an interactive visualization demo to aid with transparency and visualization of our method (Appendix K). We encourage all reviewers to explore the visualization. Additionally, we attach an anonymized version of our open-sourced codebase.

There were some questions about the baselines and generalizability of our algorithm which we thought we needed to address. At the time of submission, Moatless-tools 0.0.2 was one of the most performant open-source agents on SWE-bench. Our intention was to integrate MCTS as a stand-alone module, and make an apples-to-apples comparison with the base agent. This was a design choice made for two reasons: 1) We wanted the search algorithm to make little to no assumptions about the base agent, to ensure generalizability. 2) We wanted to avoid comparisons between different base agents, which may be hyper-optimized to maximize scores on the benchmark. Instead, we focus on the academic question of whether search would be useful or not, and under what conditions. Our code has already been open-sourced and is actively used, and we are attaching an anonymized version here. Notice that all search components are compartmentalized separately to the base agent, and interact with it via the `node.py` api. To adapt the algorithm to a new agent, it would need to be integrated with this API, and the prompts would need to be adjusted to define the different tools the particular agent is using. To create such an approach we needed to overcome notable engineering challenges.

All the top submissions on SWE-bench-lite, which is the official active benchmark suggested by the SWE-bench authors, succeed our submission, and use more recent models, particularly Claude-Sonnet-3.5. We have made a moatless-tools submission which achieves 38% (top 3) using this model. Due to this being such an active space, it is inevitable that the leaderboards will outpace the publication process. We therefore encourage the reviewers to look beyond the numbers and evaluate the merits of our efforts for what they are: introducing a fully open-source (code, models)  software agent search implementation and a case study on the key features that make it work. **Having said that, our work still stands as the SOTA approach utilizing open-source models (Qwen2.5-72B-instruct).**

Lastly, there were some questions regarding the value function. The role of the value function is to evaluate the utility of each state/action as it pertains to the particular type of state/type (search, identify, plan, edit). These criteria are defined individually in the value function prompts, which we now provide in full in Appendix G. Empirical evidence of the importance of state-specific criteria in correctly evaluating state/action types is provided in Fig. 4a. The value function is able to converge to the right solution with 73% accuracy averaged across the 5 models (Fig. 5a). We also provide further evidence of the ability of the value function to separate successful trajectories from unsuccessful ones in Appendix F, Fig. 8. The discriminator complements this approach by utilizing LLMs to directly compare the proposed solutions, improving the selection accuracy of the final solution to 84%. While this is not a standard component of traditional MCTS, we believe that it is an advantage of our method, as code changes can often be arbitrary when seen in isolation, as some reviewers also remark. The structured comparison helps identify subtle trade-offs that may not be captured by the value function alone. **As shown by the compute matching baseline requested by reviewer rRDq, the search process goes beyond brute-force to more efficiently discover a desired solution.**

We have provided the refined manuscript, demo, codebase, and updated appendix and refer to them in our responses. We would like to thank the reviewers once again for their thorough observations and suggestions which have improved our manuscript. We really appreciate your efforts.

---

> ### Author Response · Authors · 2024-11-30
> **Thanks to the reviewers.**
>
> We'd just like to conclude by thanking the reviewers and providing some final comments.
>
> ## Baselines and Random Seeds
> Regarding baselines, we note that there is a single baseline (Moatless) in our work. While additional baselines might be expected, the multiple LLMs in our study serve a role similar to random seeds, demonstrating that our results aren't a statistical fluke. The question of seeds remains challenging for us due to the significant computational expense involved, the minimal impact it would likely have on results, and notably, the fact that this type of analysis isn't present in other SWE-bench papers.
>
> ## Benchmark Requirements
> The suggestions regarding benchmarks would effectively require us to implement the algorithm with multiple agents and multiple seeds to enable apple-to-apples comparison with base agents or to outright achieve SOTA on the benchmark in order to outright claim the algorithm outperforms all submitted solutions to SWE-bench. This shifts the focus toward SOTA pursuit rather than algorithm development.
>
> ## New Results, Demo, Code
> We appreciate all the reviewers comments, but we wish they would have engaged in more insightful discussion with us. None of the reviewers acknowledged our code or interactive demo, which was meant to help them understand more about our implementation.
>
> Having said all the above, we will use your feedback to improve our work. Thanks again for all your time and efforts to review our submission.

---

### Meta-Review · Area_Chair_gjLi · 2024-12-23

**Metareview:**

This is a tricky case. This paper presents a novel and interesting solution to a very important problem with strong results. As such, it should be a clear accept, right? However, the paper as originally submitted was quite sloppily written and the method was not well explained. This has improved in response to the reviewers concerns, in particular the bibliography (although I think there other interesting things that could be in there, including attempts at using evolutionary algorithms together with LLMs for code generation). Still, I think the explanation of the method could be clearer. I also think single-run results graphs are not kosher and you should have many runs and proper statistical significance testing, unless the runs are extremely expensive - but on the other hand, the improvements are large. I also think you should compare with other methods as well, given that SWE-bench is a fairly popular benchmark.

Still, I lean towards accepting the paper, based on the originality of the method and strength of the results, and the fact that the authors have put in a lot of work towards addressing the reviewer comments.

**Additional Comments On Reviewer Discussion:**

The authors did their best to address every comment from the reviewers.

---

### Decision · Program_Chairs · 2025-01-22

Accept (Poster)